# Caspase 9-induced apoptosis enables efficient fetal cell ablation and disease modeling

Kenji Matsui [1], Masahito Watanabe[2,8], Shutaro Yamamoto [1,3], Shiho Kawagoe[1], Takumi Ikeda[1], Hinari Ohashi[1], Takafumi Kuroda[1], Nagisa Koda[1], Keita Morimoto[1], Yoshitaka Kinoshita [1,4], Yuka Inage [1,5], Yatsumu Saito[1], Shohei Fukunaga [1], Toshinari Fujimoto [1], Susumu Tajiri[1], Kei Matsumoto[1], Eiji Kobayashi [6], Takashi Yokoo [1] ✉ & Shuichiro Yamanaka [1,7] ✉

Fetal cell ablation models are crucial for studying congenital diseases, organ regeneration, and xenotransplantation. However, conventional knockout models offer limited control over disease severity, while conditional ablation models often require fetus-harming inducers. In the present study, we demonstrate that the inducible caspase 9 system enables precise targeting of fetal nephron progenitor cells in mice through the intrinsic apoptotic pathway. Using a safe, placenta-permeable inducer, this system facilitates specific, rapid, and efficient cell ablation. The system's temporal control allows precise adjustment of disease severity, generating reproducible models ranging from congenital kidney deficiency to severe chronic kidney disease. Cells with low expression levels of inducible caspase 9 and those in solid organs are less susceptible to apoptosis. However, this limitation can be overcome by inhibiting the X-linked inhibitor of apoptosis protein, thereby expanding the system's applicability. Additionally, this model provides a developmental environment suitable for chimeric kidney regeneration. This system advances understanding of induced cell death mechanisms, enhances pathological research tools, and supports therapeutic development in kidney disease and xenotransplantation applications.

Animal models that allow targeted cell or tissue ablation are powerful tools in developmental biology, regenerative medicine, and disease modeling[1–3]. Specifically, fetal cell ablation enables the study of complex mechanisms underlying organogenesis, the generation of congenital chronic disease models, and advances in organ regeneration and transplantation technologies[4–7].

We recently demonstrated that fetal kidneys transplanted into rat fetuses in utero retain urine production capability in the postnatal period, highlighting the potential of fetal intervention in treating congenital kidney diseases[8,9]. However, replicating severe congenital kidney diseases in animal models and translating these models to large animals in preclinical trials remains challenging with currently

¹Division of Nephrology and Hypertension, Department of Internal Medicine, The Jikei University School of Medicine, Tokyo, Japan. ²Meiji University International Institute for Bio-Resource Research, Kanagawa, Japan. ³Department of Urology, The Jikei University School of Medicine, Tokyo, Japan. ⁴Department of Urology, Graduate School of Medicine, The University of Tokyo, Tokyo, Japan. ⁵Department of Pediatrics, The Jikei University School of Medicine, Tokyo, Japan. ⁶Department of Kidney Regenerative Medicine, The Jikei University School of Medicine, Tokyo, Japan. ⁷Kidney Applied Regenerative Medicine, Project Research Units, The Jikei University School of Medicine, Tokyo, Japan. ⁸Present address: PorMedTec Co., Ltd., Kanagawa, Japan. ✉e-mail: tyokoo@jikei.ac.jp; shu.yamanaka@jikei.ac.jp

available technologies. Constitutive knockout models often lead to neonatal lethality due to organ defects[10]. Meanwhile, conditional knockout models using the Cre-loxP system require the maintenance of multiple transgenic lines, making them time-consuming and costly, particularly for large-animal studies[11]. Additionally, existing conditional knockout models face several limitations when applied to fetal cell ablation. For example, diphtheria toxin receptor (DTR)-based models require direct intrauterine injection of diphtheria toxin, which cannot cross the placenta, leading to increased fetal mortality[12,13]. Diphtheria toxin A (DTA) models, which depend on tamoxifen, are highly toxic and cause fetal lethality and developmental abnormalities[14,15]. Additionally, the slow induction of cell ablation in DTA models hampers precise severity control in disease modeling[16,17].

To overcome these challenges, we utilize the inducible caspase 9 (iC9) system, which triggers the intrinsic apoptotic pathway. The iC9 system consists of a modified human FK506-binding protein (FKBP–F36V) fused to human caspase 9, where the endogenous caspase activation and recruitment domain is deleted[18]. In this system, a chemical inducer of dimerization (CID), which specifically binds to FKBP–F36V and is safe for human use[19], facilitates iC9 dimerization and activation. This activation triggers the intrinsic apoptotic pathway by engaging downstream effector caspases including caspase 3, ultimately leading to cell death[20]. While the iC9 system has been shown to induce precise, rapid, and efficient cell ablation at the cellular level[21,22], its efficacy in ablating cells in adult rodent models remains suboptimal[23], with no studies reporting in vivo iC9 models of inducible organ defects. Related to this, solid tumors have been reported to exhibit lower iC9 efficiency compared to single-cell states due to unspecified resistance mechanisms[24]. Furthermore, while caspase 8, a key component of the extrinsic apoptotic pathway, has been used to target non-proliferating cells such as senescent cells and adipocytes[1,25,26], no studies have reported the removal of fetal progenitor cells via either the intrinsic or extrinsic apoptotic pathway.

In this study, we develop mouse and rat models using both knock-in and transgenic strategies for iC9, driven by the *Six2* promoter, a marker of nephron progenitor cells (NPCs)[27]. This approach enables rapid and efficient NPC-specific ablation following the administration of CID, a safe and placenta-permeable iC9 inducer. By leveraging this feature, we establish a uniform kidney disease model due to nephron loss, a key pathophysiological mechanism underlying chronic kidney disease (CKD). The severity is easily adjustable by modifying CID administration timing, ranging from neonatal lethality to severe injury developing as early as one month after birth, representing an unprecedented achievement[28]. Furthermore, the establishment of a rat fetal model, which can endure the invasiveness of transplantation surgery[8], highlights its potential for fetal transplantation therapy development. This system requires only one animal line, simplifying its applicability to larger animals. We also discover that inducing apoptosis is more challenging in settings of low iC9 expression and in solid organs than in single-cell applications. These limitations can be overcome by enhancing iC9 expression through biallelic incorporation or by co-administering an inhibitor of X-linked inhibitor of apoptosis protein (XIAP)[29,30]. In addition, we develop a Cre-loxP mouse model utilizing iC9, which, combined with these understanding of iC9 characteristics, broadens its applicability across various organs. Finally, using the NPC-deficient fetal kidneys we generate interspecies chimeric kidneys, including human–mouse chimeric kidneys that mature in vivo. These chimeric kidneys are expected to serve as an alternative therapeutic approach that combines the advantages of xenotransplantation and kidney regeneration[31–33]. Given its distinct characteristics, the iC9 system serves as a versatile CKD model with varying severities, a recipient model for severe congenital organ failure, and a platform for chimeric organ development.

## Results

### Generation of Six2-iC9 mice and in utero ablation of fetal NPCs using Six2-iC9[+/+] mice

The CAG-iC9 vector was transfected into porcine fibroblasts and mouse embryonic fibroblasts to verify the inhibition of cell proliferation by the addition of 10 nM CID to the culture medium (Supplementary Fig. 1a, b). Next, Six2-iC9 mice, which harbored iC9 and the fluorescent marker tdTomato, were generated. The selected knock-in site was immediately after the stop codon in exon 2 of *Six2* to prevent *Six2* knockout in homozygous mice, thereby averting mortality at birth resulting from frontonasal dysplasia and kidney hypoplasia (Fig. 1a)[34]. The Six2-iC9[+/−] and Six2-iC9[+/+] lines were confirmed with the PCR of genomic DNA (Supplementary Fig. 1c, Supplementary Table 1). Using primers targeting the *Six2* sequence flanking the knock-in region (*Six2* 5' and 3' arms), the knock-in allele generates a 3200-bp PCR product, which is significantly larger than the 420-bp product from the wild-type allele. In Six2-iC9[+/−] mice, amplification of the knock-in allele is inhibited; therefore, an additional PCR targeting the 5' knock-in site was performed to distinguish Six2-iC9[+/−] mice from the wild-type mice.

Knock-in fetuses were distinguished by the tdTomato expression in the frontonasal area (Fig. 1b). The kidneys of knock-in fetuses exhibited notable tdTomato expression in NPCs that formed the cap mesenchyme (CM)[27] (Fig. 1c, d). The tdTomato fluorescence was detected in both neonatal Six2-iC9[+/−] and Six2-iC9[+/+] mice, albeit with a more intense signal observed in Six2-iC9[+/+] neonates (Supplementary Fig. 1d). To evaluate CID toxicity in fetuses, 1.5 mg/kg CID was intraperitoneally administered to wild-type pregnant mice on embryonic day 13.5 (E13.5) and fetuses were naturally delivered. The fetal survival rate was 100% in both the untreated (*n* = 14) and CID-treated (*n* = 15) groups, with no change in body weight or kidney size (Supplementary Fig. 2a, b). At 2 months of age, the survival rate remained at 100% in both the untreated and CID-treated groups (*n* = 13 and 9, respectively), with no change in body weight (Supplementary Fig. 2c). In comparison, tamoxifen, the inducer used in DTA models, was orally administered at 40 mg/kg to two pregnant wild-type mice on E13.5. All newborns from both litters died, consistent with a previous report indicating that tamoxifen exerts a strong adverse effect on fetuses[15].

In mice, NPCs undergo rapid exponential proliferation, particularly between E11 and E14 whereas nephron formation continues until postnatal day 4 (P4)–P6 (Fig. 1e)[35,36]. We determined the utility of the iC9 system in arresting nephron formation by treating fetuses in utero. One dose of 1.5 mg/kg CID was intraperitoneally injected on E11.5 (when nephron formation begins), E13.5, or E15.5 (during nephron formation) to pregnant mice carrying Six2-iC9[+/+] fetuses, and the neonates were naturally delivered (Fig. 1e). The kidneys on P0.5, equivalent to E18.5, were smaller in neonates who received CID earlier in utero, with disruption observed in the tdTomato[+] CM (Fig. 1f–h). Immunostaining confirmed the disappearance of SIX2[+]/tdTomato[+] NPCs that surround cytokeratin 8 (CK8)[+] ureteric buds (Fig. 1i, j) and the reduction in NPHS1[+] mature glomeruli (Fig. 1k, l). To confirm that unexpected iC9 activation did not occur in the absence of CID, wild-type mice were compared with Six2-iC9[+/+] mice that were not administered CID. We did not observe significant differences in kidney size (Fig. 1h, Supplementary Fig. 3a), CM structure (Fig. 1j, Supplementary Fig. 3b, c), and glomerular number (Fig. 1l, Supplementary Fig. 3d) between the two groups.

### Generation of a CKD model by in utero NPC ablation

All Six2-iC9[+/+] fetuses that were administered CID on E11.5 died at birth (*n* = 3), whereas those treated on E13.5 survived, allowing postnatal evaluation. In a cohort of three litters generated via in vitro fertilization, 21 of the 32 pups (66%), including 11 males and 10 females, survived beyond 1 month, and 17 of the 32 pups (53%), including 9 males and 8 females, survived for at least 2 months. A comparison of the 1-

and 2-month old Six2-iC9⁺/⁺ pups with the wild-type male mice that were not administered CID revealed significantly higher serum creatinine (sCr), blood urea nitrogen (BUN), and urinary albumin/creatinine ratio in 1- and 2-month-old Six2-iC9⁺/⁺ mice compared to wild-type mice, except for sCr of 1-month-old female Six2-iC9⁺/⁺ mice. Urinary osmolality was significantly decreased in 1- and 2-month-old Six2-iC9⁺/⁺ mice, implying impaired urine concentrating ability (Fig. 2a). The kidneys of 2-month-old Six2-iC9⁺/⁺ mice were reduced in size and exhibited cyst formation (Fig. 2b, Supplementary Fig. 4a). There were no sex differences in these indicators of kidney injury (Fig. 2a,

Supplementary Fig. 4a). Light microscopy and immunostaining were conducted in 2-month-old male wild-type and Six2-iC9⁺/⁺ mice. A more pronounced decrease in glomerular number compared to that observed on P1.5 was observed, likely reflecting the continuation of nephrogenesis after birth (Figs. 1l, 2c, d), along with glomerular enlargement (Fig. 2e–g). The tubules were also dilated, extending from the lotus tetragonolobus lectin (LTL)⁺ proximal and E-cadherin (ECAD)⁺ distal tubules to the CK8⁺ collecting ducts in both the cortex and medulla (Fig. 2c, e, h, i), with particularly prominent findings observed in cortical collecting ducts (Fig. 2i). Furthermore, the

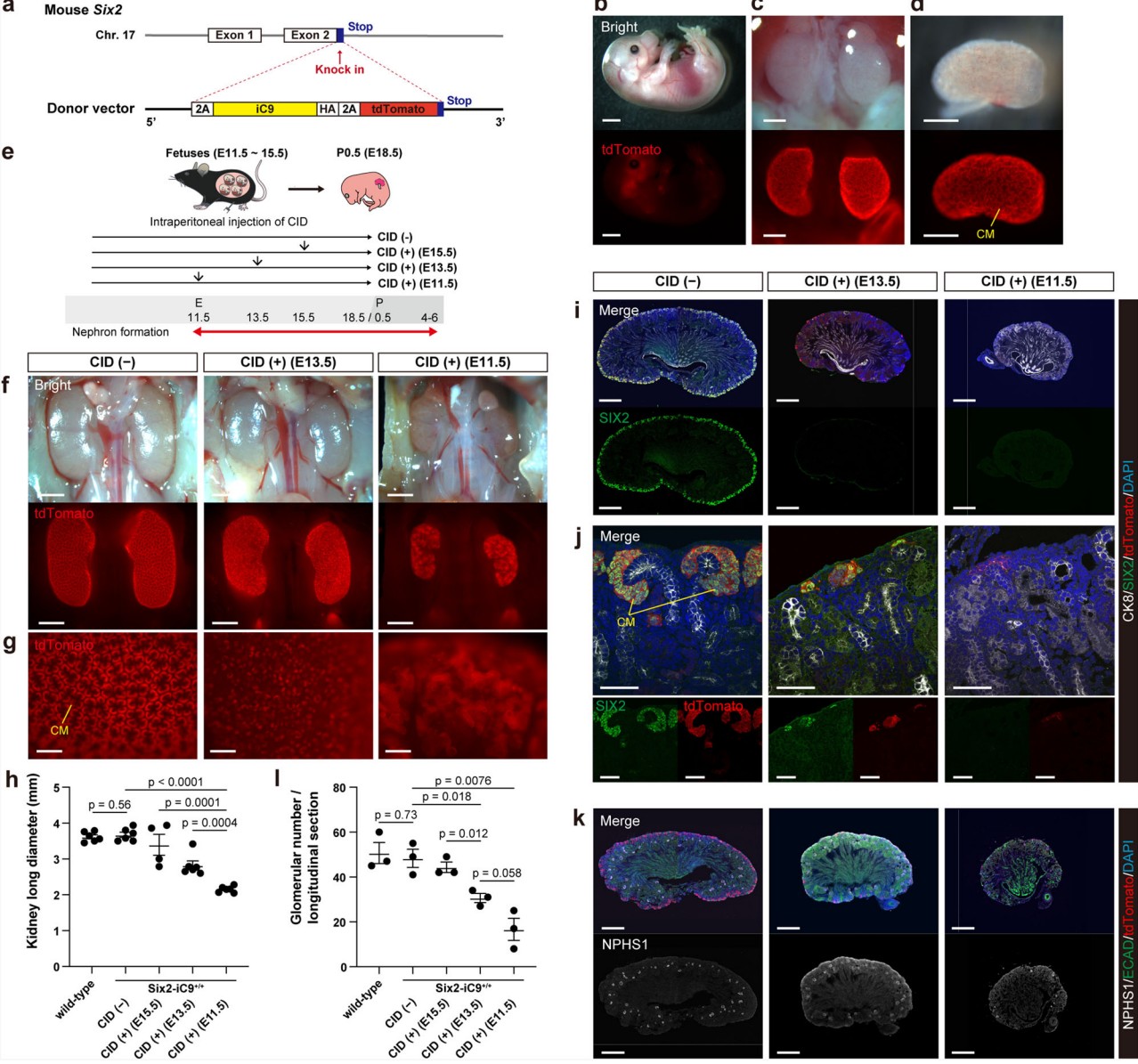

**Fig. 1 | Generation of Six2-iC9 mice and NPC ablation of Six2-iC9⁺/⁺ fetuses by intraperitoneal administration of CID to pregnant mothers. a** A schematic of the insertion site of the donor vector in the mouse *Six2* gene and the sequence of the donor vector. **b–d** Fluorescence stereomicroscopic images of fetal Six2-iC9⁺/⁺ mice on E14.5. **b** Images of a whole body. Scale bars, 2 mm. **c** Images of kidneys exposed by spinal cord removal. Scale bars, 500 μm. **d** Images of an extracted kidney. Scale bars, 500 μm. **e** A schematic of intraperitoneal injection of CID to pregnant mice carrying Six2-iC9⁺/⁺ fetuses on E11.5, 13.5, or 15.5, followed by the analysis of the offspring on P0.5 (E18.5) after natural delivery. **f** Fluorescence stereomicroscopic images of P0.5 neonatal Six2-iC9⁺/⁺ kidneys without CID administration or following CID administration on E11.5 or E13.5. Scale bars, 1 mm.

**g** Magnified images of (**f**). Scale bars, 200 μm. **h** Kidney long diameters of P0.5 neonatal wild-type and Six2-iC9⁺/⁺ with CID administration during the fetal period. **i–k** Frozen section immunostaining images of longitudinal slices of (**f**). Scale bars, 500 μm in (**i**) and (**k**), 50 μm in (**j**). **l** The glomerular number in longitudinal sections of P0.5 neonatal wild-type and Six2-iC9⁺/⁺ kidneys with CID administration during the fetal period. The data were analyzed from 6 kidneys from 3 individuals in (**h**) and from 3 individuals in (**l**) and are presented as mean ± SEM. Statistical analysis was performed using a two-tailed unpaired *t*-test. Source data are provided as a Source Data file. CID, chemical inducer of dimerization; CK8, cytokeratin 8; CM, cap mesenchyme; E, embryonic day; ECAD, E-cadherin; iC9, inducible caspase 9; P, postnatal day.

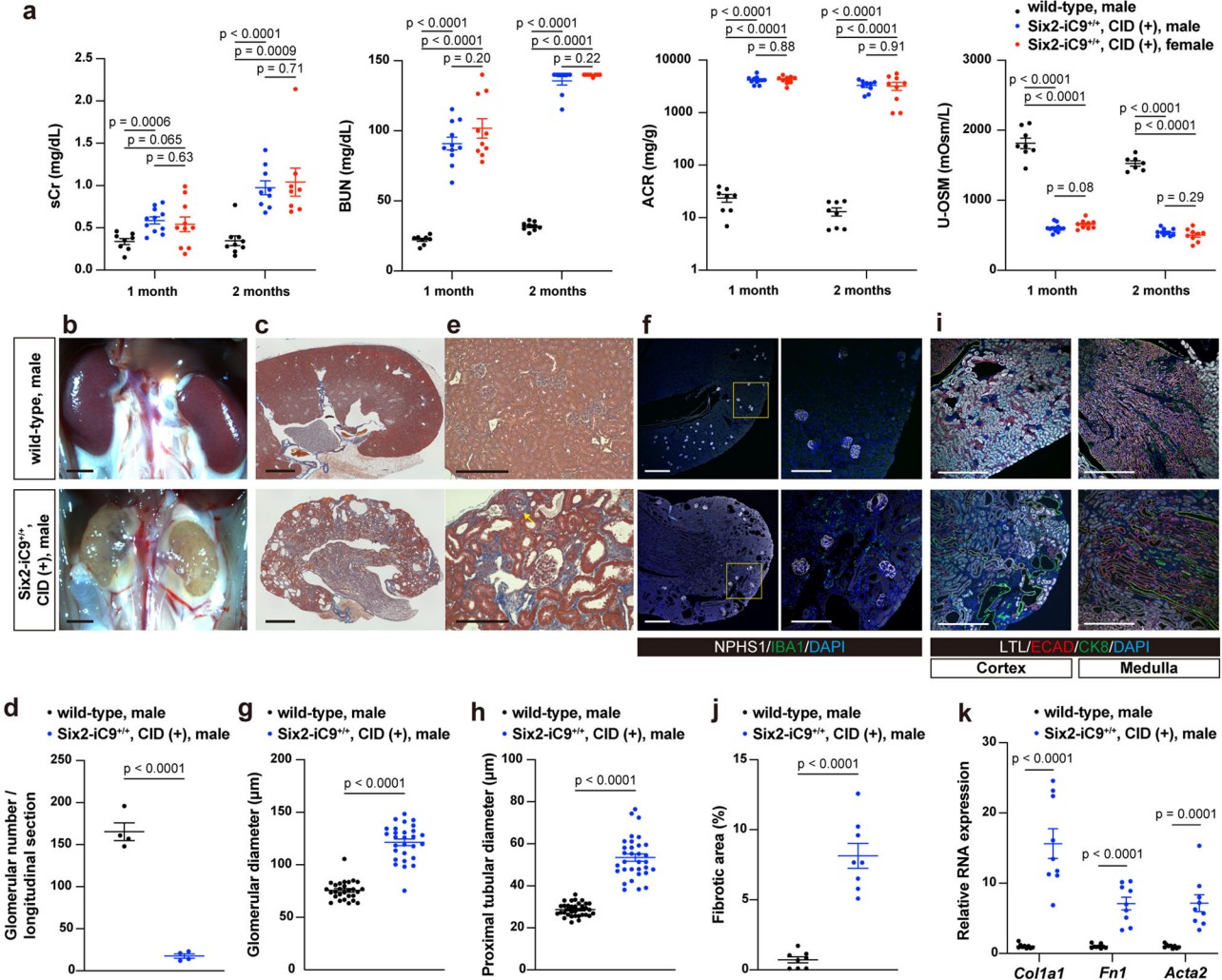

**Fig. 2 | Generation of a chronic kidney disease model by CID administration during a fetal period. a** Serum creatinine (sCr), blood urea nitrogen (BUN), urinary albumin/creatinine ratio (ACR), and urinary osmolality (U-OSM) in male and female Six2-iC9[+/+] administered CID on E13.5 at 1 and 2 months of age, compared with those of male wild-type mice. BUN measurements were only feasible for values up to 140 mg/dL. **b** Stereomicroscopic images of kidneys at 2 months of age of male wild-type mice (upper) and Six2-iC9[+/+] mice administered CID on E13.5 (lower). Scale bars, 2 mm. **c** Masson's trichrome staining of (**b**). Scale bars, 1 mm. **d** The glomerular number in longitudinal sections of kidneys at 2 months of age. **e** The magnified images of (**c**). Scale bars, 200 μm. **f** Immunostaining for IBA1[+] macrophages and NPHS1[+] glomeruli. Scale bars, 500 μm in left and 200 μm in right. Glomerular (**g**) and proximal tubular (**h**) diameter of kidneys at 2 months of age. **i** Paraffin section

immunostaining images of cortex and medulla of (**b**) that stain LTL[+] proximal tubules, ECAD[+] distal tubules, and CK8[+] collecting ducts. Scale bars, 500 μm. **j** Percentage of fibrotic areas stained blue in Masson's trichrome staining within the cortical region. **k** Relative RNA expression of fibrosis-related genes at 2 months of age. The data were analyzed using 9 male wild-type, 11 male Six2-iC9[+/+], and 10 female Six2-iC9[+/+] individuals in (**a**), 4 individuals in (**d**), (**g**), (**h**), (**j**), and 9 individuals in (**k**), and are presented as mean ± SEM. Statistical analysis was performed using a two-tailed unpaired *t*-test. Source data for all individuals used in these analyses are provided as a Source Data file. *Acta2*, actin alpha 2, smooth muscle; CID, chemical inducer of dimerization; CK8, cytokeratin 8; *Col1a1*, collagen type I alpha 1 chain; *Fn1*, fibronectin 1; IBA1, Ionized calcium binding adapter molecule 1; ECAD, E-cadherin; LTL, lotus tetragonolobus lectin.

characteristic CKD features were evident, including interstitial fibrosis (Fig. 2c, e, j, Supplementary Fig. 4b), partial tubular atrophy (Supplementary Fig. 4c), glomerular sclerosis (Fig. 2e), and infiltration of Ionized calcium binding adapter molecule 1 (IBA1)[+] macrophages (Fig. 2f). The expression levels of the fibrosis-related genes, collagen type I alpha 1 chain (*Col1a1*), fibronectin 1 (*Fn1*), and actin alpha 2, smooth muscle (*Acta2*), were elevated in male Six2-iC9[+/+] mice treated with CID (Fig. 2k). The expression levels of *fn1* and *Acta2* were significantly associated with sCr levels (Supplementary Fig. 4d).

The subcutaneous administration of 1 mg/kg CID in P1.5 Six2-iC9[+/+] mice resulted in the disruption of the tdTomato[+] CM structure (Supplementary Fig. 5a) and the elimination of SIX2[+]/tdTomato[+] NPCs (Supplementary Fig. 5b, c) by the following day (P2.5). However, at 1 month of age, these mice showed no signs of kidney injury, except for a slight increase in BUN (Supplementary Fig. 5d).

## Efficient and rapid induction of NPC ablation in Six2-iC9[+/+] fetal kidneys

Following the successful induction of a severe CKD phenotype through fetal intervention, the cellular characteristics of ablation in this model were validated using ex vivo whole-organ culture of kidneys isolated from E14.5 Six2-iC9[+/+] fetuses, which were then cultured with varying doses of CID for 4 days (Supplementary Fig. 6a). tdTomato fluorescence of NPCs decreased in kidneys treated with 100 nM CID (Fig. 3a). Immunostaining revealed that SIX2[+]/tdTomato[+] NPCs were eliminated by treatment with 100 nM CID with an efficiency of 93.6% (Fig. 3b–d). The branching of CK8[+] ureteric buds was impaired and the tip formation was absent following the NPC ablation, likely reflecting the loss of interaction with NPCs (Fig. 3b, c)[36]. The number of NPCs in Six2-iC9[+/+] kidneys cultured without CID was comparable to that in wild-type kidneys, further confirming that an unexpected

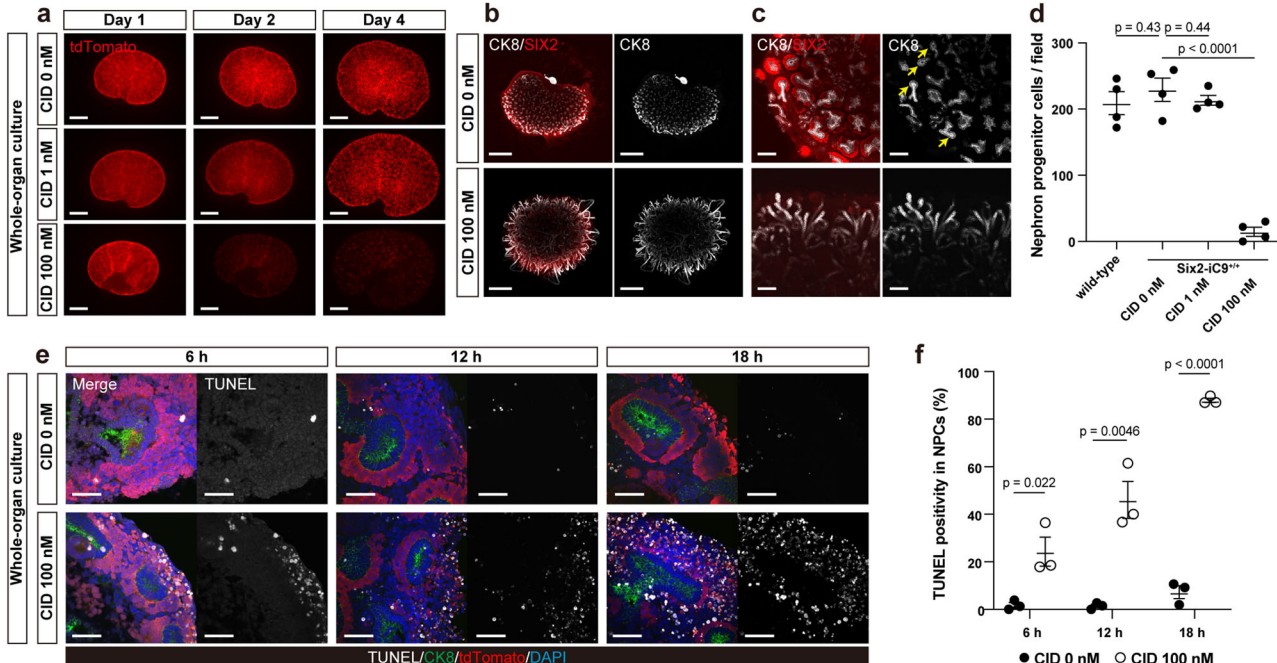

**Fig. 3 | Efficient and rapid induction of NPC ablation in homozygous fetal kidneys. a** Fluorescence stereomicroscopic images of fetal Six2-iC9+/+ kidneys cultured with 0, 1, or 100 nM CID for 4 days on the air–liquid interface. Scale bars, 500 µm. **b** Whole-mount immunostaining images of fetal Six2-iC9+/+ kidneys cultured with CID for 4 days. Scale bars, 500 µm. **c** Magnified images of (**b**). Scale bars, 100 µm. Yellow arrows indicate ureteric bud tips. **d** The number of NPCs per field of wild-type and Six2-iC9+/+ kidneys cultured with or without CID for 4 days. **e** Frozen section images of TUNEL staining along with immunostaining of fetal Six2-iC9+/+ kidneys harvested 6, 12, and 18 h after the start of culture with 0 or 100 nM CID. Scale bars, 50 µm. **f** Percentage of NPCs under apoptosis (TUNEL+/tdTomato+/DAPI+) in NPCs (tdTomato+/DAPI+) in (**e**). The data were analyzed using 4 biologically independent samples in (**d**) and 3 biologically independent samples in (**f**) and are presented as mean ± SEM. Statistical analysis was performed using a two-tailed unpaired t-test. Source data are provided as a Source Data file. CID, chemical inducer of dimerization; CK8, cytokeratin 8; TUNEL, terminal deoxynucleotidyl transferase dUTP nick end labeling.

iC9 activation did not occur in the absence of CID (Fig. 3d, Supplementary Fig. 6b).

Terminal deoxynucleotidyl transferase dUTP nick end labeling (TUNEL) staining revealed that apoptosis of tdTomato+/DAPI+ NPCs began as early as 6 hours after CID exposure, with 88% of NPCs being TUNEL-positive at 18 h (Fig. 3e, f). As a control, we evaluated the speed of apoptosis induction in Six2-DTA kidneys cultured ex vivo, in which the administration of 4-hydroxytamoxifen induces DTA expression in NPCs. The NPC ablation was observed at 48 h in kidneys treated with 2 µg/mL 4-hydroxytamoxifen, although the number of TUNEL+ NPCs was not higher than that observed in controls at 18 h (Supplementary Fig. 7a, b). The specific, efficient, and rapid removal of NPCs supported the efficacy of the intervention during the fetal period.

### Low iC9 expression below the threshold is insufficient to induce apoptosis

NPC ablation was evaluated in E14.5 Six2-iC9+/- kidneys. Unlike the same-aged Six2-iC9+/+ kidneys, no reduction in tdTomato fluorescence was observed in NPCs (Supplementary Fig. 8a) and SIX2+/tdTomato+ NPCs were not ablated (Supplementary Fig. 8b–d). Similar results were observed in conditions of higher CID concentrations of up to 5 µM or where enhanced drug penetration was induced by peeling off the renal capsule. We confirmed that CID was safe to fetal kidneys based on the absence of disruption to the CM structure (Supplementary Fig. 8b, c). Capillary western immunoassay was conducted to compare the donor vector-derived iC9 protein expression among the Six2-iC9+/+, Six2-iC9+/-, and wild-type fetal kidneys by measuring tdTomato protein levels. The tdTomato protein levels in whole kidneys of Six2-iC9+/- mice were approximately 50% lower than those of Six2-iC9+/+ mice, with β-actin used as a control (Fig. 4a, Supplementary Fig. 9a, b); similar results for

tdTomato were obtained in NPCs, with SIX2 as a control (Supplementary Fig. 9a–c). The SIX2 expression relative to β-actin was comparable among the three groups (Supplementary Fig. 9a, b, d). These results suggest that a threshold level of iC9 expression is necessary for NPC ablation[37,38] and that this threshold may lie between the homozygous and heterozygous genotypes in this model.

### Modification of cellular states by dissociation and reaggregation affects susceptibility to the induction of NPC ablation

NPC ablation was validated in dissociated and reaggregated fetal kidney cell spheres[39]. In Six2-iC9+/+ spheres, CID treatment effectively ablated the SIX2+/tdTomato+ NPCs, consistent with our observations in whole-organ cultures (Supplementary Fig. 10a, b). Surprisingly, in Six2-iC9+/- spheres, unlike that observed in whole-organ cultures, CID treatment at concentrations of 1 and 100 nM introduced at the time of reaggregation resulted in the ablation of SIX2+/tdTomato+ NPCs (Fig. 4b–d). Conversely, the treatment of spheres with 100 nM CID after initiating kidney tissue reconstruction on day 2 did not induce NPC ablation (Fig. 4d). Based on these findings, we hypothesized that the single-cell state enhances susceptibility to apoptosis. To investigate this, we compared the transcriptomes of untreated wild-type fetal kidneys collected prior to dissociation (predissociation group, n = 3) and those enzymatically and manually dissociated into single-cell suspensions (post-dissociation group, n = 3) using bulk RNA sequencing. Distinct gene expression patterns were observed between the two groups (Fig. 4e), including the upregulation of genes associated with the DNA damage response (e.g., *Egr1, Ccn1, Dusp1, Junb, Fos,* and *Btg2*) in the post-dissociation group (Fig. 4f)[40–44]. Gene ontology analysis further confirmed the enhancement of the DNA damage response in the post-dissociation group (Fig. 4g).

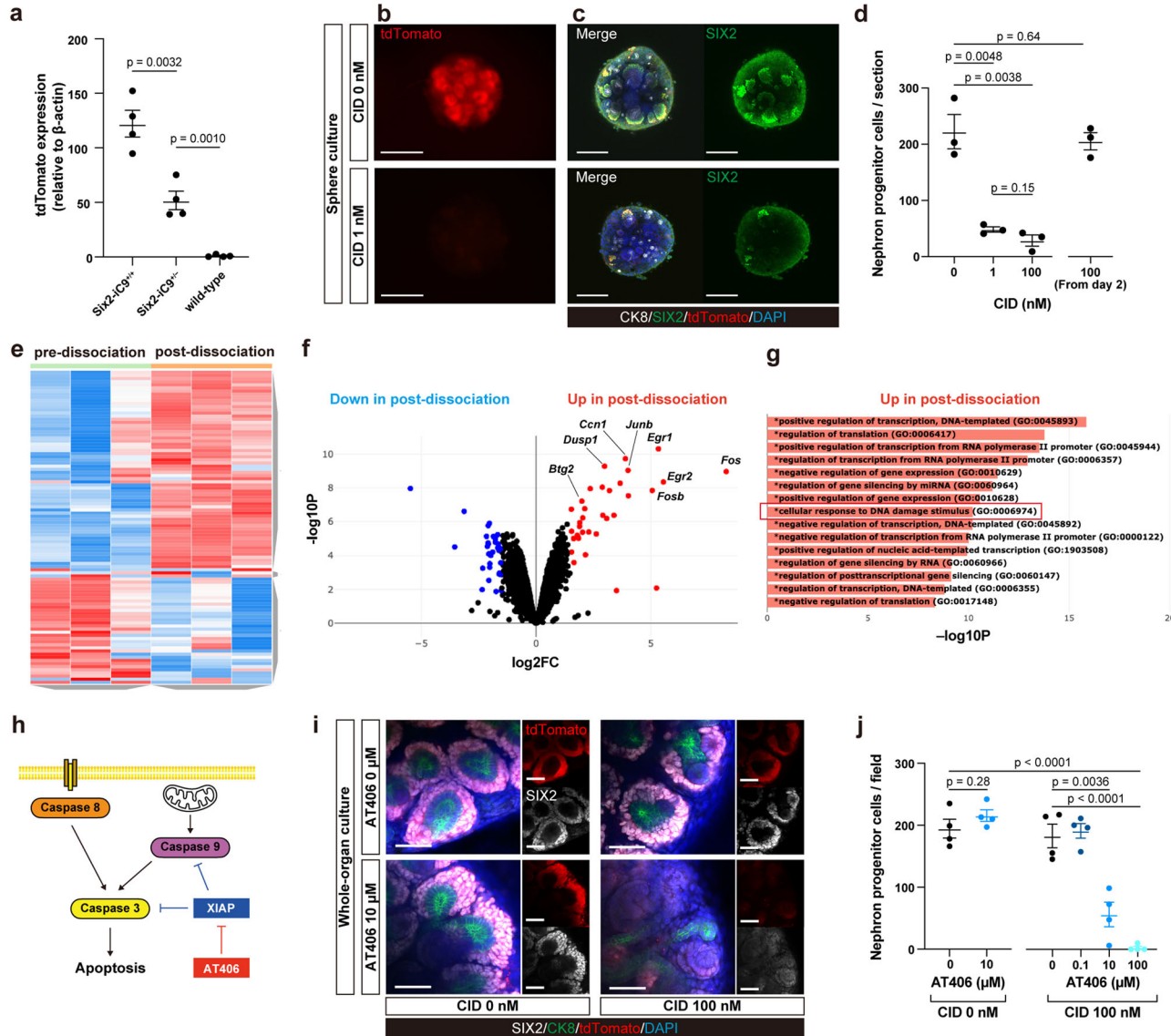

**Fig. 4 | Induction of NPC ablation in heterozygous kidneys by rendering cellular environment conducive to apoptosis. a** Relative protein expression levels of tdTomato compared to β-actin in kidneys of fetal Six2-iC9[+/+], Six2-iC9[+/−], and wild-type mice. **b** Fluorescence stereomicroscopic images of fetal Six2-iC9[+/−] kidney cell spheres cultured for 4 days without CID or with 1 nM CID added from the point of re-aggregation. Scale bars, 200 μm. **c** Whole-mount immunostaining images of (**b**). Scale bars, 100 μm. **d** The number of NPCs per field in fetal Six2-iC9[+/−] kidney cell spheres after 4 days of culture without CID, with 1 or 100 nM CID added from the point of re-aggregation to day 4, and 100 nM CID added from day 2 to 4. **e** A heatmap showing gene expression patterns of wild-type fetal kidneys without the dissociating process (pre-dissociation) and those dissociated enzymatically and manually (post-dissociation). **f** A volcano plot showing the differentially expressed genes in the comparison of "pre-dissociation" and "post-dissociation." **g** Enriched pathways from gene ontology analysis of the upregulated genes in "post-dissociation." **h** A schematic of apoptotic pathway and its association between XIAP, which is inhibited by AT406. **i** Whole-mount immunostaining images of fetal Six2-iC9[+/−] kidneys cultured with 0 or 100 nM CID and 0 or 10 μM AT406 for 4 days on the air–liquid interface. Scale bars, 50 μm. **j** The number of NPCs per field of fetal Six2-iC9[+/−] kidneys cultured for 4 days with CID and AT406. The data were analyzed using 3 biologically independent samples in (**d**) and (**e–g**), and 4 biologically independent samples in (**j**), and are presented as mean ± SEM. Statistical analysis was performed using a two-tailed unpaired *t*-test. Source data are provided as a Source Data file. CID, chemical inducer of dimerization; CK8, cytokeratin 8; XIAP, X-linked inhibitor-of-apoptosis protein.

## XIAP inhibition lowers the threshold of iC9 expression

A strategy to lower the threshold for iC9 expression would enable this system to function even under monoallelic conditions or with low-expression promoters, thereby enhancing its applicability. To this end, we utilized AT406, an XIAP inhibitor that blocks the caspases in the intrinsic apoptotic pathway (Fig. 4h)[30,45]. Fetal Six2-iC9[+/−] kidneys were cultured with and without CID and/or AT406. When used in combination with 100 nM CID, AT406 increased the efficiency of NPC ablation in a dose-dependent manner compared with the untreated controls, achieving 100% ablation at a concentration of 100 μM (Fig. 4i, j). At concentrations of up to 100 μM, AT406 alone did not induce NPC ablation or disrupt the CM structure (Fig. 4i, j); however, treatment with 500 μM AT406 alone disrupted the overall kidney structure.

To confirm the in vivo efficiency of AT406, 50 mg/kg AT406 and 0.5 mg/kg CID were subcutaneously administered to P1.5 Six2-iC9[+/−] mice, and the next-day evaluation revealed abnormal tdTomato[+] CM structures (Supplementary Fig. 11a) and the ablation of SIX2[+]/tdTomato[+] NPCs (Supplementary Fig. 11b). However, the maternal intraperitoneal administration of CID and AT406 did not result in NPC ablation in the Six2-iC9[+/−] fetuses, suggesting that AT406 does not permeate the placenta.

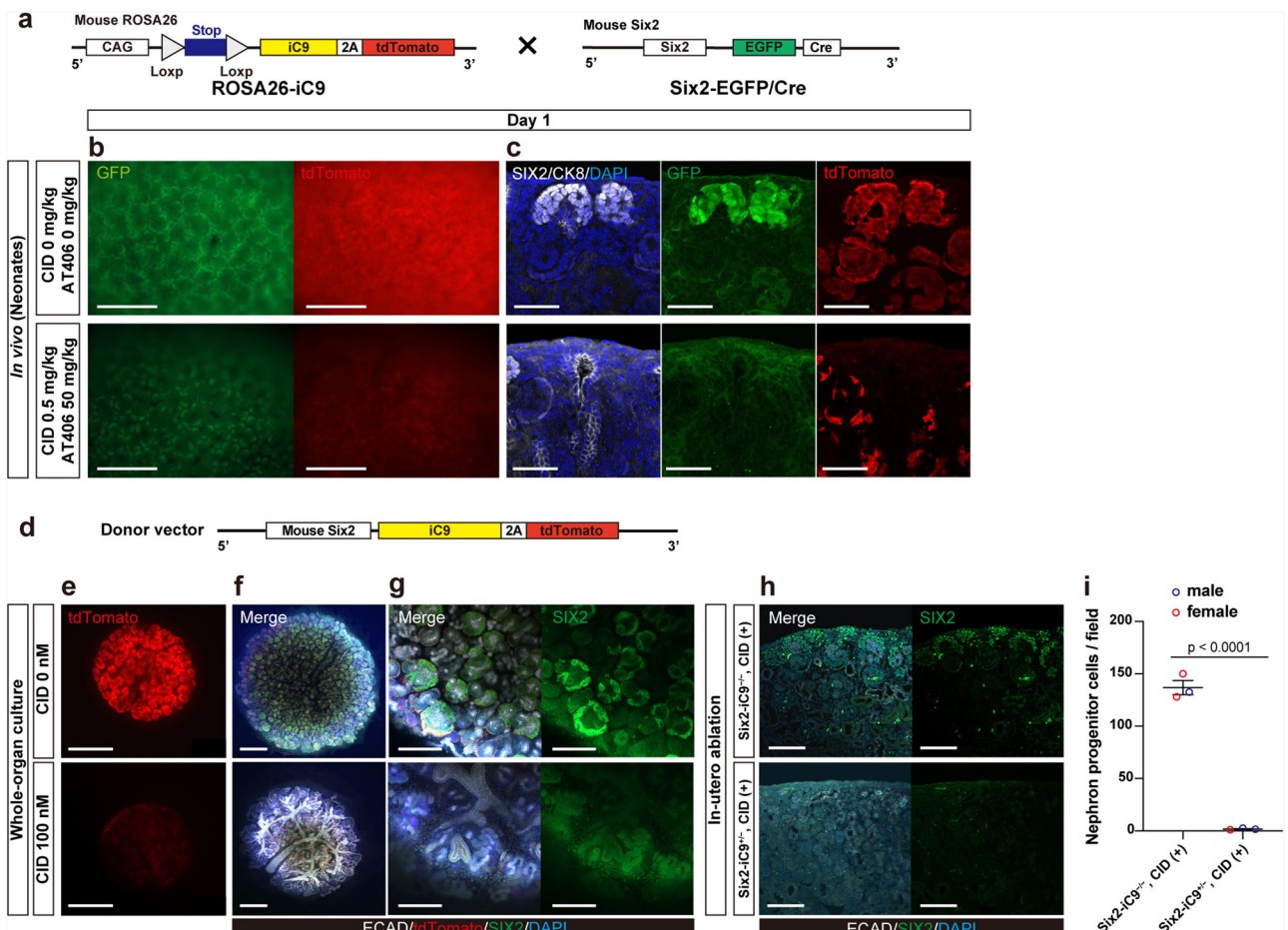

**Fig. 5 | Induction of NPC ablation in ROSA26-iC9 mice crossed with Six2-EGFP/Cre mice and in Six2-iC9 rats. a** A schematic of ROSA26-iC9 mice crossed with Six2-EGFP/Cre mice to obtain Six2-CAG-iC9 mice. **b** Fluorescence stereomicroscopic images of neonatal Six2-CAG-iC9 (P3.5) 1 day after injection of CID and AT406 into kidneys. Scale bars, 500 μm. **c** Frozen section immunostaining of (**b**). **d** A schematic of the donor vector inserted into the rat genome using bacterial artificial chromosome transgenic approach. **e** Fluorescence stereomicroscopic images of fetal Six2-iC9$^{+/-}$ rat kidneys cultured with CID on the air–liquid interface for 2 days. Scale bars, 500 μm. **f** Whole-mount immunostaining images of fetal Six2-iC9$^{+/-}$ rat kidneys cultured with CID for 4 days. Scale bars, 500 μm. **g** Magnified images of (**f**). Scale bars, 200 μm. **h** Frozen section immunostaining of fetal wild-type (Six2-iC9$^{-/-}$) and Six2-iC9$^{+/-}$ rat kidneys on E21.5 after intraperitoneal administration of CID on E17.5. Scale bars, 100 μm. **i** The number of NPCs per field of (**h**). The data were analyzed using 3 individuals per group derived from a single mother in (**i**) and are presented as mean ± SEM. Statistical analysis was performed using a two-tailed unpaired *t*-test. Source data are provided as a Source Data file. CID, chemical inducer of dimerization; CK8, cytokeratin 8; ECAD, E-cadherin; iC9, inducible caspase 9.

## Generation of mice with iC9 expression using the Cre-loxP system and rats with iC9 expression under the *Six2* promoter

To enhance the applicability of the iC9 system across various cell types more efficiently than native promoters, we generated Cre-inducible iC9 model mice (ROSA26-iC9), which express iC9 and tdTomato under the control of the *CAG* promoter in the presence of Cre. These mice were crossed with Six2-EGFP/Cre mice, which express enhanced green fluorescent protein (EGFP) and Cre in SIX2-expressing cells[27], to obtain Six2-CAG-iC9 mice (Fig. 5a). CID and AT406 were injected into the left kidneys of neonatal Six2-CAG-iC9 mice on P2.5, and the kidneys were harvested the following day. GFP and tdTomato were visible and localized to the CM without drug injection, and the signals weakened after the drug injection (Fig. 5b). Immunostaining confirmed the disappearance of SIX2$^+$/tdTomato$^+$/GFP$^+$ NPCs surrounding the CK8$^+$ ureteric buds (Fig. 5c).

Congenital organ-deficient rat models are necessary to develop fetal transplantation therapies such as solid organ transplantation. To this end, we utilized a bacterial artificial chromosome (BAC) transgenic approach in rats by inserting a donor vector containing iC9 linked to tdTomato under the mouse *Six2* gene (Fig. 5d). The culture of E15.5 rat kidneys with 100 nM CID for 4 days resulted in the successful ablation of NPCs (Fig. 5e–g). The donor vector copy number reached 30 in the analyzed line; therefore, supplementation with AT406 was not necessary even in the heterozygous kidneys. Subsequently, CID was intraperitoneally administered to pregnant rats on E17.5 and the fetuses were evaluated on E21.5. Some NPCs remained in wild-type Six2-iC9$^{-/-}$ fetuses, whereas NPC ablation was complete in the Six2-iC9$^{+/-}$ fetuses, with no sex-based differences observed (Fig. 5h, i).

## Intra- and interspecies NPC replacement and nephron regeneration with CID-induced ablation of host NPCs

We aimed to regenerate nephrons using mouse (allogeneic) and rat (xenogeneic) donor NPCs in fetal Six2-iC9 mouse kidneys by simultaneously ablating host NPCs and injecting donor NPCs. Mouse and rat renal progenitor cells (RPCs) labeled with EGFP were obtained by dissociating the kidneys of E14.5 EGFP mice and E16.5 EGFP rats. These RPCs, which contain NPCs, ureteric bud cells, and stromal progenitor cells[6], were injected under the renal capsule of the E13.5 host kidneys and cultured on the air-liquid interface (Fig. 6a, Supplementary Fig. 12a). In Six2-iC9$^{+/+}$ kidneys, 100 nM CID was added to the RPC pellets and the culture medium. In Six2-iC9$^{+/-}$ kidneys, 100 nM CID and 10 μM AT406 were added to the RPC pellets and the culture medium,

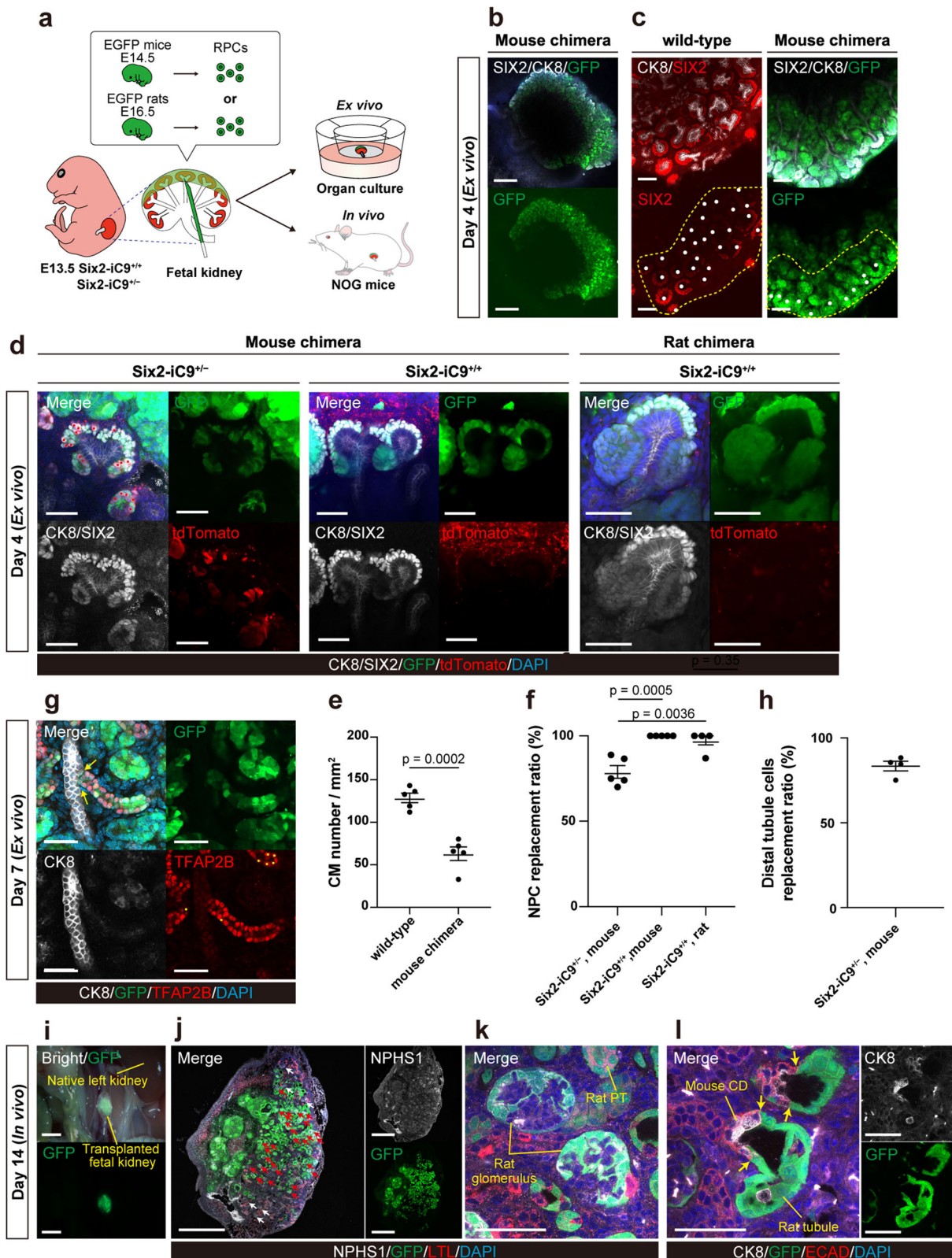

where the efficiency of NPC ablation was 71% (Fig. 4j). Four days after injection, GFP⁺ donor mouse RPCs successfully engrafted into the host fetal kidneys, in which the host tdTomato⁺ NPCs were absent (Fig. 6b). Host NPCs were replaced by the mouse or rat donor NPCs, which adhered to the host CK8⁺ ureteric buds (Fig. 6b–d, Supplementary Movies 1, 2). The host ureteric bud tips, which had been lost following NPC ablation (Fig. 3b, c), were extending into the donor NPC-derived

CMs (Fig. 6d, Supplementary Fig. 12b), suggesting the presence of an interaction between the donor NPCs and the host ureteric buds. Since the efficiency of chimera formation depends on the distribution of injected RPCs[46], we assessed the efficiency within the areas of donor cell injection rather than the entire kidney. The chimeric CMs consisting of the donor NPCs and the host ureteric buds reached 49% of the density of the native CMs (Fig. 6c, e). In the CM, the ratio of

**Fig. 6 | Replacement and maturation of nephron progenitor cells by exogenous rodent cells using fetal Six2-iC9 kidneys as a scaffold. a** A schematic illustrating the experiment. **b, c** Whole-mount immunostaining images of fetal Six2-iC9[+/−] kidneys with mouse RPC injection cultured for 4 days with 100 nM CID and 10 μM AT406, alongside fetal wild-type kidneys cultured for 4 days. White dots in (**c**) represent native CM in "wild-type" and chimeric CM in "Mouse-chimera." Scale bars, 500 μm in (**b**) and 100 μm in (**c**). **d** Whole-mount immunostaining images of Six2-iC9[+/−] kidneys with mouse RPC injection, cultured for 4 days with 100 nM CID and 10 μM AT406 (left), Six2-iC9[+/+] kidneys with mouse (middle) and rat (right) RPC injection, cultured for 4 days with 100 nM CID. Red dots represent remaining host NPCs. Scale bars, 50 μm. **e** The number of CM per mm² within the regions outlined by yellow dotted lines in (**c**). **f** Replacement ratio of NPCs in (**d**). **g** Whole-mount immunostaining images of Six2-iC9[+/−] kidneys with mouse RPC injection cultured

for 7 days with 100 nM CID and 10 μM AT406. Yellow arrows indicate the connecting points and yellow dots indicate integrated host distal tubular cells. Scale bars, 50 μm. **h** Replacement ratio of distal tubular cells by donor cells in (**g**). **i** A fluorescence stereomicroscopic image of fetal Six2-iC9[+/+] kidneys with rat RPC injection, transplanted into retroperitoneal areas of NOG mice and extracted on day 14. Scale bars, 2 mm. **j–l** Frozen section immunostaining images of (**i**). Yellow arrows in (**l**) indicate connection between rat and mouse tubules. Scale bars, 500 μm in (**j**), 50 μm in (**k**), (**l**). The data were analyzed using 5 biologically independent samples in (**e**), 5 CMs from 3 biologically independent samples in (**f**), and 4 biologically independent samples in (**h**), and are presented as mean ± SEM. Statistical analysis was performed using a two-tailed unpaired t-test. Source data are provided as a Source Data file. CD, collecting duct; CM, cap mesenchyme; NPCs, nephron progenitor cells; PT, proximal tubule; RPCs, renal progenitor cells.

replacement by allogeneic mouse RPCs was 79% in Six2-iC9[+/−] mice and 100% in Six2-iC9[+/+] mice. The ratio of replacement by xenogeneic rat RPCs was 97%, which was comparable to that observed with the allogeneic mouse RPCs (Fig. 6d, f). After extended culture, donor CMs differentiated into nephrons, including NPHS1[+]/GFP[+] glomeruli and ECAD[+]/transcription factor AP-2 beta (TFAP2B)[+]/GFP[+] distal tubules[47]. The donor distal tubules were connected with the CK8[+]/GFP[−] host collecting ducts (Fig. 6g, Supplementary Fig. 12c, d). In TFAP2B[+]/CK8[−] distal tubules, approximately 83% of the cells were GFP[+] donor cells (Fig. 6h), reflecting the contamination of CM with host NPCs (Fig. 6f).

The in vivo differentiation of regenerated rat nephrons following injection was induced by promoting vascularization. Fetal Six2-iC9[+/+] kidneys containing rat RPCs were transplanted into the retroperitoneal area of male NOD/Shi-scid, IL-2RgKO Jic (NOG) immunodeficient mice[48], which received single intraperitoneal administration of 0.5 mg/kg CID (Fig. 6a). Fourteen days post-transplantation, the GFP[+] donor cells successfully engrafted into the fetal kidney (Fig. 6i). The donor NPCs progressed into NPHS1[+]/GFP[+] mature glomeruli and accounted for 82% (28/32) of all glomeruli in the section (Fig. 6j, k). The NPCs also differentiated into proximal (LTL[+]/GFP[+]) and distal tubules, integrating with the CK8[+]/ECAD[+]/GFP[−] host collecting ducts that originated from the host ureteric buds (Fig. 6k, l).

### Nephron replacement and maturation by human NPCs via CID-induced host NPC ablation

Human NPCs were induced from GFP-labeled human induced pluripotent stem cells (hiPSCs), according to a previously reported method (Supplementary Fig. 13a, b)[49], yielding a mean of 1.38 ± 0.08 × 10⁵ cells/sphere; 66% of the cells expressed integrin subunit alpha 8 (ITGA8) but were negative for platelet-derived growth factor receptor alpha (PDGFRα) (n = 8, Supplementary Fig. 13c)[50]. Before injection, we confirmed the ability of the induced NPCs to differentiate into nephrons by observing the formation of glomeruli after 9 days in culture (Supplementary Fig. 13d–f). The obtained human NPCs were injected under the renal capsule of E13.5 Six2-iC9[+/+] kidneys and cultured with or without 100 nM CID added to the NPC pellets and the culture medium (Fig. 7a, b). In the presence of CID, the GFP[+] donor NPCs engrafted into the fetal kidneys with the ablation of the tdTomato[+] host NPCs (Fig. 7b). The culture medium contained epithelialization-promoting factors such as CHIR[51]. Therefore, the NPC replacement ratio in CMs was evaluated on day 2, earlier than the evaluation conducted when using mouse or rat RPCs (Fig. 7c, d). In the absence of CID, the injected SIX2[+]/GFP[+] human NPCs partially infiltrated the CMs as cell clusters, surrounding the CK8[+] host ureteric buds along with the SIX2[+]/tdTomato[+] host NPCs (Fig. 7c, Supplementary Movie 3). This pattern was in contrast with the scattered and sporadic incorporation of donor mouse RPCs injected into the fetal wild-type mouse kidneys[31]. Conversely, in kidneys cultured with CID, the SIX2[+]/GFP[+] human NPCs settled around the host ureteric buds while the host NPCs were ablated (Fig. 7c, Supplementary Movie 4). The NPC replacement ratio was markedly increased with the addition of CID (Fig. 7d). By day 4 of

culture, the settled human NPCs differentiated into CK8[+]/GFP[+] renal vesicle-like epithelial structures connected to the CK8[+]/GFP[−] mouse ureteric buds (Fig. 7e).

The in vivo maturation of the regenerated human nephrons was assessed by transplanting the fetal Six2-iC9[+/+] kidneys injected with human NPCs into the retroperitoneal area of the male NOG mice after 3 days of culture; the transplanted kidneys were then harvested 3 or 7 days later (Fig. 7a, f). Over time, the CD31[+]/GFP[−] mouse-derived endothelial cells infiltrated the transplanted kidney (Fig. 7g). The human NPCs differentiated into NPHS1[+]/GFP[+] glomeruli with the incorporation of mouse endothelial cells (Fig. 7h) and ECAD[+]/CK8[−]/GFP[+] distal tubules connected to the ECAD[+]/CK8[+]/GFP[−] collecting ducts originating from the fetal Six2-iC9[+/+] kidney (Fig. 7i, Supplementary Movie 5).

Finally, we evaluated the size of the components in (1) native kidneys harvested from 2-month-old mice, (2) in vivo regenerated mouse nephrons in fetal wild-type mouse kidneys collected after 7 days following transplantation into NOG mice, and (3) in vivo regenerated human nephrons in fetal Six2-iC9[+/+] kidneys, following the injection of human NPCs, which were cultured for 3 days, transplanted into NOG mice and collected after 7 days. Notably, while the native mouse glomeruli were larger than the regenerated mouse glomeruli, they were smaller than their regenerated human counterparts. Furthermore, the regenerated human glomeruli were smaller than the previously reported size of native human glomeruli (216 μm in diameter)[52] (Fig. 7h, j, Supplementary Fig. 14). Conversely, the distal tubule cells of the regenerated mouse and human nephrons were comparable in size, both being smaller than those in native mouse kidneys (Fig. 7i, k, Supplementary Fig. 14).

## Discussion

In this study, we have developed rodent models featuring a cell ablation system that employs the iC9 system, which is driven by the *Six2* promoter in mice and rats and the *CAG* promoter in mice. This system allows the rapid and efficient induction of cell death, facilitated by a placenta-permeable inducer with good safety, generating a uniform kidney disease model due to nephron loss with adjustable severity through the maternal administration of the inducer. Enhancing the expression of iC9 and using XIAP inhibition were beneficial in achieving effective targeted cell ablation. This model provides a scaffold for the development of human–mouse chimeric kidneys that mature in vivo (Supplementary Fig. 15).

One of the primary objectives in targeting fetuses for cell ablation is to create fetal organ failure models for treatment via the transplantation of xenogeneic fetal organs[8]. A preliminary study utilized recipient wild-type rats to evaluate fetal kidneys as a new therapeutic approach to treating fetuses with kidney failure[8]. Although models that can recapitulate severe kidney failure from the fetal to the postnatal stage are needed, currently available methodologies pose several limitations. Constitutive knockout models lack the ability to adjust disease severity and often result in neonatal lethality[10]. In models using

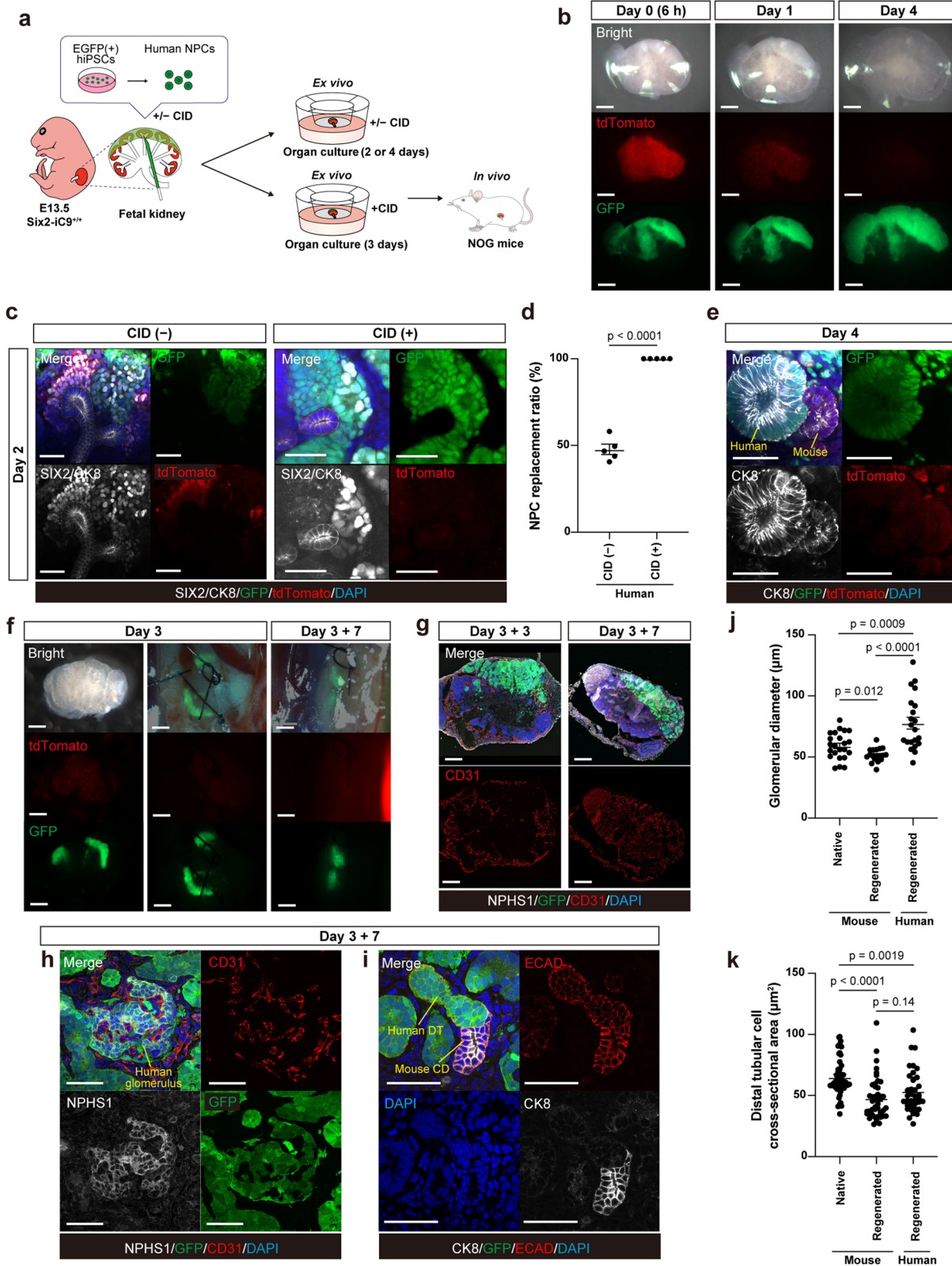

DTR, diphtheria toxin administration requires open surgery as it does not penetrate the placenta, potentially increasing neonatal mortality[12,13]. Although rodents exhibit diphtheria toxin resistance, high doses can be deleterious[53]. Models using DTA, while benefiting from the effective placental permeability of tamoxifen, are plagued by tamoxifen toxicity and high miscarriage rates, which mandates cesarean section for neonatal survival[14,15], whereas slow induction of cell

death is another limitation[16,17]. In contrast, iC9 activation by CID enables rapid and efficient NPC ablation. The good placental permeability of CID demonstrated in this study is consistent with previous reports showing its ability to cross the blood–brain barrier[54]. Together with the good safety profile of CID demonstrated in the present study, these unique characteristics render the iC9 system conducive to the creation of fetal organ failure models (Table 1). In the present study,

**Fig. 7 | Replacement and maturation of nephron progenitor cells by exogenous human cells using fetal Six2-iC9 kidneys as a scaffold. a** A schematic illustrating the experiment. **b** Fluorescence stereomicroscopic images of fetal Six2-iC9$^{+/+}$ kidneys with human NPC injection, cultured with 100 nM CID for 4 days on the air–liquid interface. Scale bars, 500 μm. **c** Whole-mount immunostaining images of Six2-iC9$^{+/+}$ kidneys with human NPC injection cultured without CID or with 100 nM CID for 2 days. Scale bars, 50 μm. **d** Replacement ratio of NPCs in the cap mesenchyme in (**c**). **e** Whole-mount immunostaining images of Six2-iC9$^{+/+}$ kidneys with human NPC injection cultured with 100 nM CID for 4 days. Scale bars, 50 μm. **f** Fluorescence stereomicroscopic images of fetal Six2-iC9$^{+/+}$ kidneys with human NPC injection, cultured with 100 nM CID for 3 days (left), transplanted to retro-peritoneal areas of male NOG mice (middle) and harvested after 7 days (right). Scale bars, 500 μm. **g–i** Frozen section immunostaining images of (**f**), harvested after 3 days (**g** left) and 7 days (**g** right, **h**, **i**). Scale bars, 200 μm in (**g**) and 50 μm in (**h**), (**i**). **j, k** Glomerular diameters (**j**) and distal tubular cell cross-sectional areas (**k**) of native mouse kidneys, regenerated mouse nephrons, and regenerated human nephrons. The data were analyzed using 5 cap mesenchymes from 3 biologically independent samples in (**d**). Analysis was conducted using 23, 18, and 21 glomeruli for (**j**) and 47, 43, and 37 distal tubular cells in (**k**) from native mouse kidneys, regenerated mouse nephrons, and regenerated human nephrons, respectively. The data are presented as mean ± SEM. Statistical analysis was performed using a two-tailed unpaired *t*-test. Source data are provided as a Source Data file. CD, collecting ducts; CID, chemical inducer of dimerization; DT, distal tubule; hiPSCs, human induced pluripotent stem cells; NPCs, nephron progenitor cells.

**Table 1 | Comparison of conditional cell ablation models for application to fetal organ failure models and to humanized chimeric organs**

|  | For the fetal organ failure model | For humanized chimeric organs |
|---|---|---|
| DTR model | • diphtheria toxin must be administered to the fetus via open surgery since it does not cross the placenta. | • It cannot be used because diphtheria toxin is toxic to human cells. |
| DTA model | • Due to delayed and incomplete cell ablation, it may be difficult to adjust the severity.<br>• Because tamoxifen has toxicity to the fetus, the mortality rate of offspring increases. | • The safety margin of tamoxifen is narrow, which may lead to damage in fetal organs containing human cells. |
| iC9 model | • The induction of cell death is rapid and efficient, enabling fine severity adjustment.<br>• The inducer is safe and placenta-permeable, enabling induction of fetal cell death through a single intraperitoneal administration.<br>• Since it does not rely on the Cre-loxP system, maintenance of one lineage is enough, possibly making it easier to apply to large animals. | • The inducer is safe for both rodent and human cells.<br>• Since it does not rely on the Cre-loxP system, maintenance of one lineage is enough, possibly making it easier to apply to large animals. |

*DTR* diphtheria toxin receptor, *DTA* diphtheria toxin A, *iC9* inducible caspase 9.

one-time administration of CID in pregnant mice between E11.5 and E15.5 resulted in complete NPC ablation in neonates. The more pronounced reduction in glomerular number and kidney size observed with earlier CID administration indicates that a single intraperitoneal injection of CID is sufficient in depleting NPCs beyond the compensatory proliferative ability of the remaining NPCs, in contrast with that observed in the DTA system[15].

Using the iC9 system, we have developed a model of CKD with precise and adjustable nephron loss, reproducing uniform kidney damage across all animals with no sex-dependent differences. Nephron loss is a common pathophysiologic feature associated with low birthweight, aging, and hypertension, which imposes hemodynamic and metabolic overload on remaining nephrons[55]. The imposed burden placed by nephron loss stimulates signaling including PPARα, AMPK, and mTORC1, leading to tubular hypertrophy and cyst formation, thereby contributing to CKD progression[56]. Indeed, we observed the characteristic CKD features, including glomerular hypertrophy and sclerosis, tubular dilation and atrophy, interstitial fibrosis, and inflammatory cell infiltration, in this model[57]. Traditional animal models of CKD often rely on non-physiologic insults, such as nephrotoxic substances, renal parenchymal resection, and unilateral ureteral obstruction[58,59]. The existing models of low nephron numbers are typically based on low birth weight caused by factors such as maternal malnutrition[60] and premature birth[61], which impact the entire fetal body, and controlling disease severity is challenging. We recently used the DTR model to generate an NPC ablation-induced CKD model by administering diphtheria toxin through the placenta. Despite intervening at the same developmental stage used in the present study (E13.5), the resulting damage was much milder with the DTR model than with the iC9 model; furthermore, the phenotypes emerged only with a high-salt diet[62]. Our findings demonstrate the advantages of the iC9 model over existing models, including the ability to induce severe and uniform CKD through physiologic nephron loss at as early as 1 month of age via a single systemic maternal drug administration. The detailed mechanisms underlying the progression from nephron loss to

kidney injury remain incompletely understood, and our model should serve as a valuable experimental tool for future investigations on potential therapeutic targets.

We also successfully generated rats expressing iC9 under the control of the mouse *Six2* promoter using the BAC transgenic technology[63]. We adopted this approach for several reasons. First, the donor vector copy numbers are higher in transgenic animals than in knock-in animals, which is an advantage given that higher iC9 expression levels are beneficial for apoptosis induction. Up to 30 copies were present in the rats analyzed in the present study. Second, transgenic lines can be established more quickly compared with the knock-in approach. Finally, validation in Six2-iC9 mice had already confirmed that the mouse *Six2* promoter is capable of sufficiently driving iC9 expression. Rats are more valuable as disease models because of the relatively more straightforward acquisition of physiologic and biochemical information[64]. Rat fetuses can also endure the invasiveness of transplantation surgery, rendering them suitable for fetal transplantation models[8]. Future studies may utilize the iC9 system with the porcine knock-in technology[65]. Incorporating endogenous promoters such as *Six2* facilitates lineage maintenance in a single strain, which is simpler than the Cre-loxP system.

The previous lack of highly effective animal models utilizing iC9 might be attributed to resistance to apoptosis induction. Although studies have utilized iC9 in transfection and knock-in experiments[24,37,38], detailed in vivo validation has not been previously performed. First, our comparison of the homozygous and heterozygous fetal kidneys revealed a threshold for iC9 expression levels, consistent with in vitro findings[37,38]. Second, we found that cell ablation in dissociated and reaggregated fetal kidney spheres was inducible, even in heterozygous animals exhibiting low iC9 expression, indicating that inducing cell ablation might be more challenging in solid organs than in single-cell states, consistent with previous studies demonstrating that cell death could not be effectively induced after solid tumor formation[24]. It is unlikely that the enzymatic and physical

dissociation induces iC9 expression or inhibits gene silencing, which is considered to contribute to resistance to cell death[37]. Therefore, we compared the transcriptomes of fetal kidney cells before and after dissociation to identify factors associated with susceptibility to apoptosis and found marked increases in the expression levels of DNA damage response genes following dissociation, including *Egr1, Ccn1, Dusp1, Junb, Fos*, and *Btg2*[40–44]. DNA damage induces the intrinsic apoptotic pathway via mitochondria[66]. In addition, the single-cell state promotes cell death in a caspase-dependent and caspase-independent manner by activating actomyosin through the Rho-ROCK pathway[67,68], with increases in caspase 9 and caspase 3[67]. Thus, apoptosis may be more easily induced in single cells compared to solid organs. The comparison of solid organs and dissociated and reaggregated spheres may serve as a valuable research tool to elucidate the mechanisms of apoptosis. Third, methods other than cell dissociation should be considered to promote apoptosis in vivo in the setting of low iC9 expression levels. We found that cotreatment with CID and the XIAP inhibitor AT406 promoted apoptosis in heterozygous kidneys. XIAP inhibits caspase 9 as well as its downstream effectors caspase 3 and caspase 7[29,30]. XIAP expression is related to the efficiency of iC9-induced apoptosis[45]. XIAP directly inhibits the dimerization of caspase 9[69]; thus, AT406 might be unique to the intrinsic apoptotic pathway. Taken together with the establishment of iC9 in the Cre-loxP model, this system may be applicable to other organs, such as pancreas and heart, to create various fetal organ failure models[70,71].

Alternative therapies are sorely needed for end-stage kidney disease due to the shortage of kidneys available for transplantation. Although various methods have evaluated the generation of ex vivo kidney organoids to mimic kidney development[49,72,73], these organoids do not connect to the urinary tract after transplantation and are unable to survive long-term[74]. Conversely, recent efforts to mitigate immune rejection utilizing multiple gene edits have led to improvements in xenotransplantation[75,76]. We propose fetal organ complementation, a concept that combines kidney regeneration with xenotransplantation[31–33], as an alternative approach. Regenerated human nephrons can connect with the scaffold mouse nephrons[32,51,77], and the present study demonstrates that human nephrons connected to mouse collecting ducts can be regenerated in vivo. Furthermore, if transplanted with an intact connection to the ureter and bladder, fetal kidneys can be anastomosed with the recipient's urinary tract[78]. Additionally, partially replacing cells in fetal organs with allogeneic components can reduce rejection[33]. Blastocyst complementation, another approach for generating chimeras using pluripotent stem cells, faces challenges due to evolutionary disparities that act as barriers[79] and ethical concerns regarding the potential development of humanized nervous and reproductive systems[80]. Recent attempts to integrate human skin into fetal mice at later developmental stages to circumvent interspecies barriers align with our strategy of creating organogenesis-stage chimeras[7].

The DTR model[31] is not useful in kidney replacement using human NPCs, because of the inherent presence of DTR in human cells. Although previous studies used the in vitro tamoxifen-inducible DTA model to demonstrate human–mouse cell interactions in relatively immature nephrons[32], this model raises concerns due to the narrow safety margin of tamoxifen[14]. We hypothesized that the iC9 system, which ensures safe, rapid, and efficient NPC removal, could provide a scaffold for the formation of human–mouse chimeric nephrons (Table 1). Our findings demonstrate the in vivo formation of mature human nephrons connected to mouse tubules.

The debate surrounding the size of donor-derived organs in chimeras involves various factors associated with each species[81,82], which may pose challenges in chimera formation. Notably, the regenerated human glomeruli were larger than the mouse glomeruli, implying the presence of intracellular, species-specific, size-regulating mechanisms. Conversely, no significant variations were observed in the size of tubular cells across the species, indicating that size differences had a minimal impact on interspecies cell adhesion within the tubules.

The present study has several limitations. First, AT406 was administered within a short period in neonates and we did not thoroughly evaluate its systemic toxicity and off-target effects. Additionally, AT406 is not placenta-permeable. Future efforts should include the exploration of placenta-permeable drugs that can promote iC9-mediated apoptosis induction by intervening in other apoptotic pathways. Second, the rate of contribution and the maturation of chimeric nephrons and their long-term functional evaluation should be further improved. Considering that the density of regenerated CMs was about half of that of the native CMs even when utilizing allogeneic cells, improvements in the cell injection method are necessary to achieve maximum replacement[46]. Additionally, the developmental trajectories of xenogeneic nephrons should be evaluated to improve the rate of replacement by human NPCs. Regarding species-specific differences, cellular size, developmental velocities, and paracrine factors should be evaluated. It is also required to compare the speed of NPC elimination between the iC9 and the DTA models, to determine its impact on replacement efficiency. Third, it remains unclear whether adult organs, especially those with robust compensatory proliferation such as the liver[4], can be targeted for ablation. We aim to verify this aspect in models that use other promoters or in those created by breeding with the ROSA26-iC9 mice with the appropriate Cre mouse lines. As demonstrated in the present study, increasing iC9 expression and the combined use of XIAP inhibition is a promising approach to increase ablation efficacy. Furthermore, combining genetic engineering strategies, such as selecting appropriate promoters, utilizing enhancers, or employing innovative systems like combinatorial protein dimerization to sustain high gene expression while minimizing leakage[83], can be explored.

In summary, we demonstrate that iC9 enables efficient and safe fetal cell ablation, generating highly reproducible kidney disease models with adjustable nephron loss, spanning from congenital kidney deficiency to viable severe CKD. XIAP inhibition enhances iC9 efficacy, expanding its potential applications across diverse organs utilizing Cre-loxP and other promoter systems. This approach can also provide controlled developmental environments for chimeric kidneys. The high efficiency of iC9 in fetal cell ablation offers valuable insights into developmental processes, advancing both our understanding of the pathophysiology underlying CKD progression and the development of therapeutic strategies, including xenotransplantation approaches.

## Methods

### Ethics statement
Animals were treated in accordance with the Guidelines for Proper Conduct of Animal Experiments. All experimental procedures were approved by the animal ethics committee of the Jikei University School of Medicine (approval numbers: 2020-039 and D2021-007).

### Research animals
Six2-iC9$^{+/-}$ mice were established with a background of NOD-Sirp-α congenic C57BL/6-*Rag2$^{null}$ IL2rγ$^{null}$* [48]. Mouse Six2-targeted Platinum TALEN mRNAs were constructed using the Platinum Gate TALEN Kit (Addgene Kit #1000000043)[84] and were injected into pronuclear-stage embryos along with the donor vector (mouse Six2 arm-P2A-iC9-P2A-tdTomato). Subsequently, 2-cell stage embryos were transplanted to pseudopregnant mice. Founder mice were selected by genomic PCR and the line was established. Six2-iC9$^{+/+}$ mice were obtained by crossing Six2-iC9$^{+/-}$ with each other. ROSA26-iC9 mice were established with a background of C57BL/6JJcl. Donor vector (*CAG* promoter-Loxp-STOP-Loxp-iC9-P2A-tdTomato) was injected into pronuclear-stage embryos along with Cas 9 protein and guide RNA to target mouse Rosa26 locus. Subsequently, 2-cell stage embryos were transplanted to pseudopregnant mice. Founder mice were selected by genomic PCR

and the line was established. Six2-iC9 rats were established with a background of Sprague-Dawley rats by incorporating a donor vector, which includes iC9 and tdTomato downstream of mouse *Six2*, into the host genome using a BAC transgenic approach. Pregnant female C57BL/6JJms (B6) mice, C57BL/6-Tg (CAG-EGFP) mice (EGFP mice), Sprague-Dawley-Tg (CAG-EGFP) rats (EGFP rats), adult male and female B6 mice, and adult female ICR mice, were purchased from SLC. Adult male NOG mice were purchased from CLEA and used at 8–16 weeks of age. B6;129-Gt(ROSA)26Sor[tm1(DTA) Mrc]/J mice (ROSA26-DTA mice), B6;129-Six2[tm3(EGFP/cre/ERT2) Amc]/J mice (Six2-EGFP/CreERT2 mice), and Tg(Six2-EGFP/cre)1Amc/J (Six2-EGFP/Cre mice) were purchased from Jackson Laboratory. Male and female Six2-iC9$^{+/+}$ mice were crossed with each other to obtain Six2-iC9$^{+/+}$ offspring and with female or male B6 mice to obtain Six2-iC9$^{+/-}$ offspring. Six2-EGFP/CreERT2 mice were crossed with ROSA26-DTA mice to obtain Six2-DTA fetuses. Six2-EGFP/Cre mice were crossed with ROSA26-iC9 mice to obtain Six2-CAG-iC9 mice. Timed mating was employed for breeding: 12:00 PM on the day vaginal plug was detected was considered E0.5. All animals were maintained on a 12-hour light/dark cycle with free access to standard diet and water.

### Validation of the capacity of iC9 to induce apoptosis using porcine fetal fibroblasts

A primary culture of porcine fetal fibroblasts (male, Large White/Landrace × Duroc) was cultured in MEM α (12561-056, Gibco) supplemented with 15% fetal bovine serum (FBS, SH30070.03, HyClone Laboratories, Inc.) and divided into the following groups and cultured on type I collagen-coated 6-well plates (AGC Techno Glass) at a density of $1 \times 10^5$ cells per well: No vector with electroporation pulses (pulse voltage, 1100 V; pulse width, 30 ms; and pulse number 1), cultured with 10 nM CID (635058, Takara Bio Inc.); CAG-iC9 vector based on pMSCV-F-del Casp9.IRES.GFP vector (Addgene plasmid # 15567) transfection by electroporation pulses, cultured without CID; and CAG-iC9 vector transfection by electroporation pulses, cultured with 10 nM CID. Electroporation was conducted according to manufacturer's instructions using the Neon Transfection System (Thermo Fisher Scientific). The cell proliferation was evaluated by the percentage of cell coverage on the bottom of the dish using a 72-hour time-lapse.

### Generation of congenital CKD mice and their blood and urea evaluation

To evaluate the toxicity of CID, B6 pregnant mice were intraperitoneally administered with 1.5 mg/kg CID on E13.5. The offspring were born via natural delivery and observed for two months. To evaluate the toxicity of tamoxifen, pregnant mice carrying Six2-DTA fetuses were orally administered with tamoxifen E13.5. The birthrate of the offspring was evaluated. To generate CKD models, pregnant mice carrying Six2-iC9$^{+/+}$ fetuses were intraperitoneally administered with 1.5 mg/kg CID on E11.5, 13.5, or E15.5. The offspring were born via natural delivery, and their kidneys were harvested after euthanization on the day of birth (P0.5). To generate adult CKD model mice by administering CID on E13, in vitro fertilization was performed using sperm and oocytes from Six2-iC9$^{+/+}$ mice. The resulting embryos were transferred into female ICR mice, followed by intraperitoneal CID injection on E13, and natural delivery was allowed. For blood test, the 1- and 2-month-old offspring were anesthetized by isoflurane (2817774, Pfizer) inhalation and their facial vein was punctured with an animal lancet (18310400, MEDI-point), to collect their blood using a hematocrit capillary (2-454-21, AS ONE Corporation). The capillaries were sealed with wax (2-454-22, AS ONE Corporation) and centrifuged at $12,000 \times g$ for 10 min to separate the serum and blood cells. sCr and BUN levels were measured using a DRI-CHEM (FUJIFILM). For urinalysis, the 1- and 2-month-old offspring were placed in a mouse metabolic cage (KN-645, Natsume Seisakusho Co., Ltd.) overnight. Urine was collected in 1.5-mL tubes and

centrifuged at $4000 \times g$ for 15 min at 4 °C, and the supernatant was submitted to SRL, Inc. to measure urinary creatinine, albumin, and osmolality. Kidneys from 2-month-old mice were subjected to evaluation of pathology and RNA expression.

### Whole-organ culture of isolated fetal kidneys

The fetal kidneys were extracted from E13.5–14.5 Six2-iC9$^{+/+}$ or Six2-iC9$^{+/-}$ mice using micro-tweezers (11253-25, Dumont) and moved onto the air–liquid interface of a polycarbonate filter with an average pore size of 0.4 µm (3401, Corning) using wide pore tips (2069 G, Molecular Bio Products). They were cultured in rodent basal medium (MEM α supplemented with 20% FBS and 1% penicillin/streptomycin [168-23191, Wako]), supplemented with CID up to 5000 nM, and/or AT406 (S2754, Selleck Chemicals) up to 100 µM. The kidney structure and tdTomato fluorescence were observed over time using a fluorescence stereomicroscope (M205FA, Leica Microsystems). The fetal kidneys of Six2-DTA mice were cultured in rodent basal medium supplemented with 2 µg/mL 4OHT. The kidneys were cultured at 37 °C in a 5% $CO_2$ atmosphere, with the medium replaced every other day until collection on day 4 for evaluation via immunostaining.

### Drug administration to neonatal mice

CID with or without AT406 was administered subcutaneously or under renal capsule to neonates, using a 34 G Hamilton syringe (Saito Medical Instruments Inc.) under general anesthesia by isoflurane. The neonatal mice were then returned to their cages with their biological mothers and euthanized by decapitation the following day, after which their kidneys were extracted.

### Culture of dissociated and re-aggregated fetal kidney cell spheres

The fetal kidneys were extracted from E13.5–14.5 Six2-iC9$^{+/+}$ or Six2-iC9$^{+/-}$ mice and collected into a 1.5-mL tube containing 1 mL of accutase (Innovative Cell Technologies), and vortexed for 30 s. They were incubated at 37 °C for 15 min, with vortex after 5 min and gentle manual pipetting using a 200-µL pipette tip after 10 and 15 min, then centrifuged at $300 \times g$ for 5 min. Pellets were resuspended in 1 mL rodent basal medium and dissociated by gentle manual pipetting. Cell suspensions were passed through a 40-µm cell strainer (BD Falcon). The suspension density was adjusted to $1 \times 10^6$ cells/mL and $2 \times 10^5$ cells were distributed in each well of a U-bottom 96-well low-cell binding plate (Thermo Fisher Scientific) supplemented with CID up to 100 nM. Finally, the plate was centrifuged at 1000 rpm for 4 min and incubated overnight at 37 °C to form re-aggregates. The medium was replaced every other day and samples were collected on day 4 for evaluation via immunostaining. For some samples, exposure to CID 100 nM was started on day 2.

### Dissociation of EGFP-labeled mouse and rat fetal kidneys

Fetal kidneys from EGFP mice (E14.5) or EGFP rats (E16.5) were dissociated for injection to obtain RPCs, as described above. Rodent basal medium containing cell suspension was supplemented with 100 nM CID for injection into Six2-iC9$^{+/+}$ fetal kidneys and with 100 nM CID and 10 µM AT406 for injection into Six2-iC9$^{+/-}$ fetal kidneys. Subsequently, the suspension was centrifuged at $700 \times g$ for 3 min. The supernatant was removed completely, and the tubes were tapped to mix the pellet and incubated on ice for up to 2 h before use.

### Human induced pluripotent stem cells maintenance

GFP-labeled hiPSCs (317-12-Ff)[85] were maintained on iMatrix-511 (Nippi)-coated 6-well plate (IWAKI) in StemFit AK02N medium (Ajinomoto). Cells were cultured at 37 °C in a humidified atmosphere of 5% $CO_2$ and passaged every 7 days using TryPLE Select (Thermo Fisher Scientific). The medium was replaced on days 2, 4, and 5.

## Induction of human NPCs from hiPSCs and dissociation of NPC spheres

NPCs were induced from hiPSCs according to established methods[49]. On day 0, 10,000 iPSCs per well were re-aggregated in a V-bottom, 96-well, low-cell binding plate (Sumitomo Bakelite) that contained hNPC basal medium (DMEM/F12 [11320-033, Gibco] supplemented with 1% GlutaMAX (100X) [35050-61, Gibco], 1% Insulin-Transferrin-Selenium [41400-045, Gibco], 1% MEM Non-Essential Amino Acids Solution (100X) [11140-050, Gibco], 90 uM 2-Mercaptoethanol [21985-023, Gibco], and 0.5% penicillin/streptomycin) supplemented with 1 ng/mL human activin A (R&D Systems), 20 ng/mL human basic fibroblast growth factor (b-FGF, R&D Systems), and 10 µM Y27632 (Wako). On day 1, the spheres were transferred to a U-bottom, 96-well, low-cell-binding plate that contained hNPC basal medium supplemented with 10 µM CHIR (Axon Medchem) and 10 µM Y27632. Every other day thereafter (days 3 and 5), half of the medium was replaced with the medium of the same composition as day 1. On day 7, the spheres were transferred to a U-bottom, 96-well, low-cell-binding plate that contained hNPC basal medium supplemented with 3 ng/mL human bone morphogenetic protein 4 (R&D Systems), 0.1 µM retinoic acid (Sigma-Aldrich), 10 ng/mL human activin A, 3 µM CHIR, and 10 µM Y27632. On day 9, the spheres were transferred to a U-bottom, 96-well, low-cell-binding plate that contained hNPC basal medium supplemented with 5 ng/mL human glia activating factor 9, 1 µM CHIR, and 10 µM Y27632. On day 12 or 13, the spheres were enzymatically dissociated in 0.5 × TrypLE™ Select (Thermo Fisher Scientific) and dissociated into cell suspensions. To eliminate any remaining clumps, the cells were passed through a 40-µm cell strainer and centrifuged at $300 \times g$ for 3 min. After 100 nM of CID was added to some suspensions and the supernatant was removed, the tubes were tapped to mix the pellet, incubated on ice, and used within 2 hours.

## Flow cytometry of human NPCs

To evaluate the purity of NPCs in day 12 NPC spheres, flow cytometry was performed. Biotinylated anti-ITGA8 (R&D Systems), allophycocyanin-conjugated streptavidin (BioLegend), and phycoerythrin-conjugated anti-PDGFRα (BioLegend) were used for cell staining. The allophycocyanin (ITGA8)-positive and phycoerythrin (PDGFRα)-negative fraction was regarded as the NPC population. The cut-offs were determined using a negative control without the primary antibodies. A minimum of $5 \times 10^5$ cells was secured in each experiment.

## Ex vivo differentiation of human NPCs

To check the differentiation capacity of NPCs from each induction, day 12 NPC spheres were transferred onto Transwell and cultured in KR5 medium (DMEM/F12 [(1:1) (1×), Thermo Fisher Scientific] with 5% KnockOut™ Serum Replacement [Thermo Fisher Scientific]) supplemented with 3 µM CHIR and 200 ng/mL b-FGF for 2 days. The medium was changed to KR5 medium on day 2 and then changed every other day. The spheres were collected on day 9 and evaluated in hematoxylin−eosin (HE) staining.

## Cell injection into the subcapsular legion of fetal Six2-iC9+/+ or Six2-iC9+/− mice kidneys

Fetal kidneys of Six2-iC9+/+ or Six2-iC9+/− mice on E13.5 were exposed by removing spinal cords in a 10-cm dish containing Hank's Balanced Salt Solution. Each cell pellet of EGFP-labeled mouse or rat fetal kidney cells, or human NPCs prepared as described above, was aspirated in a glass needle (G-100, Narishige) that was previously sharpened using a pipette puller (PC-10, Narishige). The needle was inserted through the renal hilus to the capsule and cells were injected slowly by mouth pipetting with a 1-mL syringe and an aspirator tube assembly (2-040-000, Drummond Scientific Company) connected to the glass needle, under a stereomicroscope. Approximately $4.0 \times 10^5$ cells per one kidney were injected.

## Ex vivo and in vivo maturation of chimeric kidneys

For ex vivo maturation, the injected kidneys were extracted from the fetuses and placed onto the air−liquid interface. Six2-iC9+/+ kidneys with injection of mouse or rat fetal kidney cells were cultured in rodent basal medium supplemented with 100 nM CID, and Six2-iC9+/− kidneys with injection of mouse fetal kidney cells were cultured in rodent basal medium supplemented with 100 nM CID and 10 µM AT406. 0/+ kidneys with injection of hNPCs were cultured in hNPC basal medium supplemented with 1 uM CHIR, 5 ng/ml fibroblast growth factor 9 (FGF9), 10 µM Y27632, with or without 100 nM CID. The following day, the medium was changed to hNPC basal medium supplemented with 10 ng/ml FGF9, with or without 100 nM CID. The samples were collected on day 2 or 4 for evaluation via immunostaining. For in vivo maturation, Six2-iC9+/+ kidneys with injection of rat fetal kidney cells were transplanted to male NOG mice soon after the injection. The recipient NOG mice were anesthetized using isoflurane inhalation and a laparotomy was performed through an abdominal midline incision. A pocket was created in the retroperitoneum using micro-tweezers under a fluorescence stereomicroscope. One or two fetal kidneys per recipient were transplanted into each pocket, which was then closed with 8-0 nylon thread (Muranaka Medical Instruments Co. Ltd.). The operation was completed by closing the abdomen and the recipient NOG mice received single intraperitoneal administration of 0.5 mg/kg CID. Six2-iC9+/+ kidneys with injection of hNPCs were cultured for 3 days as described above, detached from the air−liquid interface, and then transplanted to male NOG mice without CID administration to the NOG mice.

## Whole-mount immunostaining

Cultured fetal kidneys and spheres were fixed with 4% paraformaldehyde (PFA, 161-20141, Wako) for 15 min at 4 °C, washed three times in phosphate buffered saline (PBS). Samples were blocked using 1% donkey serum, 0.2% skimmed milk, and 0.3% Triton X/PBS for 1 hour at room temperature, and incubated overnight at 4 °C with primary antibodies (Supplementary Table 2). After washing three times with PBS, the samples were incubated with secondary antibodies conjugated with Alexa Fluor 488, 546, or 647, and 4′,6-diamidino-2-phenylindole (DAPI) for 1 hour at room temperature. Samples were mounted with a glycerol-based liquid mountant (S36972, Invitrogen™). Images were obtained using a laser confocal microscope (LSM880, Carl Zeiss).

## HE staining and immunostaining of frozen sections

Fetal kidneys, kidneys of neonatal congenital CKD mice, in vivo matured chimeric kidneys, and ex vivo differentiated hNPC spheres, were fixed with 4% PFA overnight and dehydrated in 15% sucrose in PBS overnight and in 30% sucrose in PBS overnight at 4 °C. Specimens were embedded in an OCT compound (Sakura Finetek), and 10 µm thick-frozen sections were prepared. HE staining was performed according to the standard procedure. Antigen retrieval for immunostaining was performed in HistoVT One (Nacalai Tesque) in a warm bath at 70 °C for 20 min. After blocking with Blocking One Histo (Nacalai Tesque) for 10 min at room temperature, the sections were incubated with primary antibodies (Supplementary Table 2) and then with secondary antibodies conjugated with Alexa Fluor 488, 546, or 647, and DAPI for 1 hour at room temperature. Sections were mounted with a glycerol-based liquid mountant. TUNEL assay for apoptosis detection (C10619, Invitrogen) was performed on sections of cultured fetal kidneys following the manufacturer's protocol. Images were obtained using a microscope (BZ-X800, KEYENCE) and a laser confocal microscope.

## Masson's trichrome staining, periodic acid-Schiff staining, and immunostaining of paraffin sections

Kidneys of adult congenital CKD mice were fixed in 4% PFA, embedded in a paraffin block, and 4-µm sections were prepared. Masson's

trichrome staining and periodic acid-schiff (PAS) staining were performed according to the standard procedures. For immunostaining, the slides were deparaffinized and incubated with citrate buffer (K035, Cosmo Bio Co., Ltd.) at 121 °C for 10 min for antigen retrieval, followed by blocking and incubation with primary and secondary antibodies as described previously.

### Measurement of the number of remaining NPCs in cultured fetal kidneys

Whole-mount immunostaining images of cultured fetal kidneys were acquired with a laser confocal microscope. The number of NPCs was counted manually in one randomly selected section in peripheral areas (using ×20 lens, 694 μm square) from each kidney by two investigators.

### Measurement of the number of remaining NPCs in cultured fetal kidney spheres

Whole-mount immunostaining images of cultured fetal kidney spheres were acquired with a laser confocal microscope, focusing on the middle depth of the sphere. The number of all NPCs in the image was counted manually by two investigators.

### Measurement of the percentage of NPCs under apoptosis in cultured fetal kidneys

Frozen section images of TUNEL staining along with immunostaining of cultured fetal kidneys were acquired with a laser confocal microscope. The numbers of NPCs and TUNEL-positive NPCs were counted manually in one randomly selected section in peripheral areas (using ×63 lens, 225 μm square) from each kidney by two investigators.

### Measurement of the glomerular number in neonatal kidneys

Frozen section immunostaining images of longitudinal slices of the middle of neonatal kidneys were acquired with a laser confocal microscope. The number of NPHS1-positive structures was counted manually from each kidney by two investigators.

### Measurement of the glomerular number and diameters of glomeruli and proximal tubules in adult kidneys

PAS staining images of the middle of cross-sections of adult kidneys were acquired with a microscope. The number of glomeruli were counted manually from each kidney. Glomerulus area (S) of glomeruli was measured using Image J. Diameter (2r) was calculated as follows, estimating glomeruli are circles: $S = \pi r^2$. Outer diameter of the cortical proximal tubules was measured using Image J. Data collection was conducted by two investigators.

### Measurement of the fibrotic areas in adult kidneys

Masson's trichrome staining images of the middle of cross-sections of adult kidneys were acquired with a microscope. The fibrotic areas that are stained blue were measured using Image J (using ×20 lens, 630 μm square).

### Measurement of the NPC replacement ratio in the cap mesenchyme by donor NPCs

Whole-mount immunostaining images of cultured fetal kidneys with injection of mouse or rat RPCs, or human NPCs, were acquired with a laser confocal microscope. The numbers of SIX2$^+$/GFP$^+$ donor NPCs and SIX2$^+$/GFP$^-$ host NPCs were counted using ×20 lens manually by two investigators.

### Measurement of glomerular diameter and distal tubule cell cross-sectional area

Frozen section immunostaining images were acquired with a laser confocal microscope from the following samples: native mouse kidneys in 2-month-old mice; regenerated mouse nephrons in fetal wild-type mouse kidneys transplanted to NOG mice and harvested after 7 days; and regenerated human nephrons in fetal Six2-iC9$^{+/+}$ kidneys with human NPC injection, cultured for 3 days, transplanted to NOG mice, and harvested after 7 days. Glomerulus area (S) of NPHS1-positive round structures was measured using Image J. Diameter (2r) was calculated as follows, estimating glomeruli are circles: $S = \pi r^2$. Distal tubule cell cross-sectional area of ECAD-positive tubules was measured using Image J. Data collection was conducted by two investigators.

### Evaluation of protein expression levels

The kidneys from 2–4 E14.5 fetuses from the same mother were pooled as one biological replicate and immediately frozen at −80 °C. After thawing, 100 μL Lysis buffer with protease inhibitor and phosphatase inhibitor (2332336, ATTO) was added to each sample. The samples were grinded using a BioMasher (320103, Nippi) and then homogenized using an ultrasonic homogenizer (UR-20P, TOMY SEIKO Co., Ltd.) for 3 min on ice. After incubation on ice for 20 min, the tubes were centrifuged at 16,750 rpm for 10 min at 4 °C. The supernatant was collected into a new tube and protein concentration was measured using Pierce™ BCA Protein Assay Kits (23225, Thermo Fisher Scientific) and a microplate reader (SH-1000 Lab, Hitachi High-Tech). The sample was to a protein concentration of 0.25 mg/ml and subjected to capillary Western immunoassay (Wes, ProteinSimple) using primary antibodies listed in Supplementary Table 3, according to the manufacturer's protocol.

### RNA-seq for fetal kidneys before and after dissociation

We dissected E14.5 fetal kidneys from B6 mice. For "pre-dissociation," we collected 3 kidneys per sample, and for "post-dissociation," we extracted over $1 \times 10^6$ cells with viability of over 80% from eight kidneys per sample using the aforementioned cell dissociation method. RNA extraction and DNA removal were then performed following the protocol with the RNeasy Plus Micro Kit (74034, QIAGEN). We confirmed that each sample yielded at least 66 ng of extracted RNA, which was stored at −80 °C until the following analysis. Reverse transcription was carried out using the SMART-Seq v4 Ultra Low Input RNA kit (634892, Takara Bio), followed by library preparation using the Nextera XT DNA Library Prep Kit (FC131-1024 and FC-131-1096, Illumina). Sequencing was performed using Novaseq X plus (Illumina). The obtained data were mapped to the mouse genome using GENCODE/GRCm39 (mm39) as a reference and normalized for each gene's expression to obtain TPM. Subsequent analyses were performed using BioJupies[86].

### RT-qPCR for kidneys

Quick-DNA/RNA Miniprep Plus Kit (D7003, Zymo Research) was used to isolate RNA from kidneys at 2 months of age. RNA was reverse transcribed into cDNA using the PrimeScriptTM RT Reagent Kit with gDNA Eraser (Takara). qPCR was performed using TaqManTM Gene Expression Master Mix (Thermo Fisher Scientific), TaqMan Gene Expression Assays (Supplementary Table 4, Qiagen), and a thermocycler (Rotor-Gene Q, Qiagen). Glyceraldehyde 3-dehydrogenase was used as a housekeeping gene to normalize the expression levels.

### Statistics and reproducibility

No statistical method was used to predetermine sample size. No data were excluded from the analyses. The administration of CID and AT406 was randomly assigned to fetal kidneys. The investigators were not blinded to allocation during experiments and outcome assessment. All experiments were independently repeated at least three times with similar results, and representative fluorescence stereomicroscopy and immunostaining images are shown in the figures. All data are presented as means ± S.E.M. Data were analyzed using the two-tailed unpaired $t$-test. A $P$ value of 0.05 was considered statistically

significant. Experimental data were analyzed using GraphPad Prism software, version 8.0 (GraphPad Software), and Microsoft Excel (Microsoft).

## Reporting summary

Further information on research design is available in the Nature Portfolio Reporting Summary linked to this article.

## Data availability

All sequencing data generated in this study have been deposited in Gene Expression Omnibus (GEO) with the accession number GSE282139. The raw data used in all figures in this study are provided in the Source Data. Source data are provided with this paper.

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

## Acknowledgements

We thank H.Hayashi, M.Tamatsukuri, M.Mine, and H.Kim for experimental/technical assistance, M.Goto at Central Institute for experimental animals for support in generating Six2-iC9 and ROSA26-iC9 mice

and in vitro fertilization, Y.Totsuka and T.Saito at Institute of Immunology Co., Ltd. for support in generating Six2-iC9 rats, LAIMAN, Inc (https://laiman.co.jp/) for the creation of Supplementary Fig. 15, and Enago (https://www.enago.jp/) for the English language review. T.F. declares support for this research from JSPS-KAKENHI [JP23K15255]. K.Matsumoto declares support from JSPS-KAKENHI [grant number JP23K07725], Kidney Research Initiative-Japan: Japan Kidney Association and Nippon Boehringer Ingelheim Joint Research Project, and Eli Lilly Japan innovation research grants 2023. T.Y. declares support from AMED [grant number JP24bm1223003] and JSPS-KAKENHI [grant number JP21K08288]. S.Yamanaka declares support from AMED [grant numbers JP22bm0704049 and JP24bm1123036], JST FOREST Program [grant number JPMJFR2011], and JSPS-KAKENHI [grant numbers JP19K17756 and JP24K11439].

## Author contributions

K. Matsui, M.W., and S. Yamanaka designed the experiments and wrote the manuscript. K. Matsui, M.W., S. Yamamoto, S.K., T.I., H.O., T.K., and S. Yamanaka carried out the experiments and analyzed the data. N.K., K.Morimoto, Y.K., Y.I., Y.S., S.F., T.F., S.T., and K.Matsumoto interpreted the data and revised the manuscript. K.Matsumoto, T.Y., and S.Yamanaka acquired funding. E.K., T.Y., and S.Yamanaka supervised the project. All authors have approved the final version of the manuscript.

## Competing interests

The authors declare no competing interests.
