## [Transparent Peer Review file · Nature Communications]

Caspase 9-induced apoptosis enables efficient fetal cell ablation and disease modeling

Corresponding Author: Dr Shuichiro Yamanaka

Version 1:

Reviewer comments:

Reviewer #1

(Remarks to the Author)

In the manuscript, Matsui et al. established Six2-iC9 mice to ablate nephron progenitor cells using inducer of dimerization (CID). In Six2-iC9 het mice, inhibition of XIAP enhanced NPC ablation. The authors also established a Cre-dependent Rosa26-CAG-iC9 conditional knock-in mouse line, which is potentially useful for those who are interested in the iC9 system in vivo. Lastly, Six2-iC9 embryonic mouse kidneys injected with mouse/rat RPCs or human NPCs were cultured ex vivo or transplanted into immunodeficient mice with CID. These mouse/rat RPCs or human NPCs could differentiate into nephrons in the Six2-iC9 embryonic mouse kidneys.

The authors already claimed that Six2CreERT2; R26RDTA system resulted in efficient ablation of nephron progenitor cells and that human iPSC-derived NPCs could be successfully tested in the Six2CreERT2; R26RDTA system in their publication 4 years ago (Cell Reports 2020). It is also known that transplantation of NPCs supports maturation of nephrons for a decade (Taguchi et al, Cell Stem Cell 2014).

The significance and advancement in the current study is very rudimentary. The manuscript should be published elsewhere.

Major points:

1. From whole kidney views in Fig 6b, 6c, 6g, 7b, 7g, it is very unlikely that injected GFP+ cells interact with Six2-iC9 tissues extensively. The injected GFP+ cells merely seem to be proliferating around injection sites and these dividing cells might be occasionally incorporated into the CD of host mice. Did the injected GFP+ RPCs/NPCs enhanced UB branching of host kidney? Is there any other evidence for reconstitution of tissue interactions between GFP+ cells and host tissues, other than a few images of GFP+ cells next to host CK8+ cells in Fig 6i and Fig 7i?
2. How many per a kidney did the authors identify the connections of GFP+ cells and the host mouse CD in Fig 6i and Fig 7i when how many nephrons (or glomeruli) are formed? The numbers of connections and glomeruli per a kidney should be presented. Because many nephrons are formed, these GFP+DT to CD connections should be observed frequently if GFP+ cells interact with the host mouse kidney.
3. Do those GFP+ cells adjacent to host CK8+ cells express DT markers? ECAD is not specific to the DT. A DT-specific marker should be used replacing ECAD.
4. Was there any occasion where GFP+ cells are incorporated adjacent to CK8- non-CD host cells? If so, the number of the occasions per a kidney should be presented.
5. NPC can differentiate when cultured ex vivo or transplanted into immunodeficient mice. What is significant advantage with the use of Six2-iC9 kidneys? Appropriate controls should be used in the study.
6. The authors already claimed that Six2CreERT2; R26RDTA system resulted in efficient ablation of nephron progenitor cells and that human iPSC-derived NPCs could be successfully tested in the Six2CreERT2; R26RDTA system (Cell Reports 2020). Some minor differences were presented in Supplementary Fig. 4a, b. However, for testing human NPCs, it was unclear what was the advancement in the iC9, compared with the published Six2CreER-iDTR.

Reviewer #2

(Remarks to the Author)

In this manuscript, the authors utilized the iC9 strategy to generate mouse and rat models, enabling specific and efficient elimination of nephron progenitor cells (NPCs). Taking advantage of this feature, the researchers established a severe fetal renal failure model, which allows postnatal survival and adjustment of disease severity based on the timing of inducer administration. The study revealed that apoptosis is more challenging at lower expression levels of iC9 and in solid organ states, and the issue can be addressed by enhancing iC9 expression. Additionally, the researchers successfully developed an effective Cre-loxP mouse model using Caspase 9-induced apoptosis system, which holds potential for applications in various organs. By utilizing NPC-depleted fetal kidneys, the study also generated intra-species and inter-species chimeric kidneys, contributing to advancements in regenerative medicine research. The unique characteristics of the iC9 model make it a valuable receptor model for severe congenital organ failure and can be used to create models of chronic diseases with varying degrees of severity and for studying organ chimerism. The result is exciting although Caspase 9-induced apoptosis system is mentioned previously. Overall, the experiments were meticulously designed, and the manuscript is well-written. However, following issues should be addressed before it is accepted for publication. s

1. In Result 1, the authors aimed to generate Six2-iC9 mice. However, it is unclear why chose to test the CAG-iC9 vector in porcine fibroblasts instead of conducting relevant experiments in mouse fibroblasts. It is recommended that relevant experiments should be performed in mouse cells to provide a more coherent progression.
2. In Supplementary Fig. 1b, it is essential for the authors to offer a comprehensive explanation regarding the presence of a band around 318 bp in the wild type sample and the absence of a 3200 bp band in the Six2-iC9^{+/-} (heterozygous) sample. A detailed clarification needs to be provided.
3. In Supplementary Fig. 2, the authors assessed the toxicity of CID on fetuses by evaluating the survival rate of E18.5 fetuses through cesarean section. For a more rigorous assessment, it would be beneficial to examine the postnatal survival rate of the mice after birth.
4. In Fig. 2a, the samples size of biological replicates for measuring Serum creatinine (sCr), blood urea nitrogen (BUN), and urine albumin-to-creatinine ratio (ACR) in 1 month or 2 months is inconsistent. Can the authors elucidate this inconsistency? The sample size (n) should be provided in the corresponding figure legends.
5. Do clinical phenotypes of chronic kidney disease (CKD) exhibit cyst formation, a decrease in glomerular numbers, or cortical proximal tubule dilatation? To highlight the significance of utilizing the ablation system for constructing animal disease models, a comprehensive and detailed comparison between the CKD model and clinical phenotypes should be provided, elucidating the similarities and differences between them and emphasizing the importance and implications of the model for studying CKD.
6. Fig 2i showed immunostaining of Six2-iC9 mice treated with CID. Proper controls, including Six2-iC9 mice without CID treatment and wild-type mice, should be performed.
7. There is a lack of wild-type controls in Fig. 1i, 1j, and Fig. 3. The expression level of Six2 can provide insights about whether the Caspase 9-induced apoptosis system is due to leakage. Therefore, it is crucial to conduct a detailed evaluation of Six2 expression levels in both Six2-iC9 mice with or without CID treatment and wild-type mice.
8. The authors stated that the addition of the XIAP inhibitor lowers the threshold level of iC9 expression. The extent of threshold reduction should be quantified, for example, by comparing the level of NPC ablation in Six2-iC9^{+/+} mice treated with CID only and Six2-iC9^{+/-} mice treated with both AT406 and CID.
9. The iC9 system used in the study has not been improved compared to previous versions. As a result, the efficiency of cell removal in adult rodent models still remains unsatisfactory and unavoidable. How can you ensure the efficient achievement of complete organ ablation? To achieve more efficient whole-organ ablation, optimization can be explored in terms of dimerization-based ablation systems (<https://doi.org/10.1038/s41589-023-01281-x>). Based on this, a brief discussion can be conducted.
10. The results in line 221-224 is inconsistent with the figures. The corresponding Fig. 4b, 4c, and 4d should actually refer to Fig. 4h, 4i, and 4j, respectively.
11. The authors employed a bacterial artificial chromosome (BAC) transgenic approach to introduce a donor vector containing iC9 linked with tdTomato under the mouse Six2 gene. Why not use the rat's Six2 promoter? In addition, targeted gene knock-in can provide more stable integration and expression of the desired transgene. Why not insert iC9 linked with tdTomato specifically after the last exon of the rat's Six2 gene? Please provide necessary explanation for this.
12. In Figure 6g-i, the authors indicate rat cell integration to fetal Six2-iC9^{+/+} kidney by immunohistochemistry. The chimeric efficiency should be provided using flow cytometry.

Reviewer #3

(Remarks to the Author)

The study by Matsui and colleagues demonstrates that the inducible caspase 9 (iC9) system allows targeted ablation of mouse fetal nephron progenitor cells leading to defective kidney development and CKD of variable severity. The severity of phenotypes can be modified by the timing of induction whereas comparison of homozygous and heterozygous phenotypes reveals that a threshold level of iC9 is needed to achieve NPC ablation. Furthermore, they show that single cell state increases susceptibility to apoptosis and the addition of XIAP inhibitor lowers the threshold of cell ablation. Because of their interest to apply this system in fetal transplantation studies, they generated Six2-Cre-CAG-iC9 mice Six2-iC9 rats and used them for proof-of-concept studies on intraspecies and interspecies nephron regeneration.

Critique

This is a beautiful study with important implications in the field of kidney regeneration. The new models developed, and their included specific applications are interesting and provide novel insights. However, another potential application as a CKD model to test novel therapeutics is somewhat overlooked and would deserve more emphasis. Here are my comments

1. Have the authors assessed how consistent was the ablation of NPCs and the reduction of mature glomeruli within the CID exposed fetuses? The graphs in Fig 1 h and i show 3-5 mice per group whereas an average litter should be closer to 8 mice. Have they assessed whether mouse gender affected the degree of ablation? The reviewer has similar concerns for Figure 5.
2. Could the authors provide more details in the variability of the CKD model described in Figure 2? While they started with 18 mice (mentioned in results), they report the functional measurements for 5-6 mice per group. The pathology analysis is also limited to 3 mice. It would be important to include the phenotypes of the entire cohort to get a better idea of the potential variability.
3. To better characterize the CKD phenotype, it would be helpful to include RT-PCR analysis of pro-fibrotic genes. Was there inflammatory cell infiltration along with the fibrosis?
4. Since CID administration in Six2-iC9 neonates leads to ablation of NPCs (SuppFig 5), have the authors examined if these mice develop CKD? Probably there is a typo in the results, Six2-iC9^{+/-} instead of Six2-iC9^{+/+}. Likewise, the legend of Supp Fig 6 mentions homozygous instead of heterozygous.

Reviewer #4

(Remarks to the Author)

Version 2:

Reviewer comments:

Reviewer #1

(Remarks to the Author)

There is no response addressing my concern regarding the significance or novelty of the current study. It has been well established over the past decade that transplantation of kidney organoids promotes maturation, which is not a novel finding of this study. For years, the authors have already demonstrated that chimeric fetal kidneys with NPC are effective and useful.

Moreover, the advancement claimed in the current study is not definitive. To substantiate the claimed progress over their previous publications, the authors should include a comparison using DTA as a negative control. Without proper side-by-side controls, no definitive conclusions can be drawn. Addressing this issue requires extensive experimentation rather than mere editorial changes or minor experiments.

Reviewer #2

(Remarks to the Author)

The authors have conducted additional experiments and I have no further requests for the paper.

Reviewer #3

(Remarks to the Author)

The authors have sufficiently addressed my concerns

Reviewer #4

(Remarks to the Author)

Point-by-point Responses to the Reviewers' Comments

Date: January 2, 2025

Journal: *Nature Communications*

Manuscript ID: NCOMMS-24-21286A-Z

Manuscript title: Caspase 9-induced apoptosis enables efficient fetal cell ablation and disease modeling

Authors: Kenji Matsui, Masahito Watanabe, Shutaro Yamamoto, Shiho Kawagoe, Takumi Ikeda, Hinari Ohashi, Takafumi Kuroda, Nagisa Koda, Keita Morimoto, Yoshitaka Kinoshita, Yuka Inage, Yatsumu Saito, Shohei Fukunaga, Toshinari Fujimoto, Susumu Tajiri, Kei Matsumoto, Eiji Kobayashi, Takashi Yokoo and Shuichiro Yamanaka

Responses to the reviewers' comments

We greatly appreciate the reviewers' detailed comments and suggestions. We have revised the manuscript and the figures to address these points. We believe that they have been improved through these revisions. Below are our point-by-point responses to the comments. Responses to the comments are shown in red in the revised manuscript and in this letter.

Reviewer #1

In the manuscript, Matsui et al. established Six2-iC9 mice to ablate nephron progenitor cells using inducer of dimerization (CID). In Six2-iC9 het mice, inhibition of XIAP enhanced NPC ablation. The authors also established a Cre-dependent Rosa26-CAG-iC9 conditional knock-in mouse line, which is potentially useful for those who are interested in the iC9 system in vivo. Lastly, Six2-iC9 embryonic mouse kidneys injected with mouse/rat RPCs or human NPCs were cultured ex vivo or transplanted into immunodeficient mice with CID. These mouse/rat RPCs or human NPCs could differentiate into nephrons in the Six2-iC9 embryonic mouse kidneys. The authors already claimed that Six2CreERT2; R26RDTA system resulted in efficient ablation of nephron progenitor cells and that human iPSC-derived NPCs could be successfully tested in the Six2CreERT2; R26RDTA system in their publication 4 years ago (Cell Reports 2020). It is also known that transplantation of NPCs supports maturation of nephrons for a decade (Taguchi et al, Cell Stem Cell 2014).

The significance and advancement in the current study is very rudimentary. The manuscript should be published elsewhere.

We thank the Reviewer for their important feedback and apologize for the confusion in delineating the significance and novelty of our model.

The significance and advancement of our model lie in the use of the intrinsic apoptotic pathway to target and remove fetal cells with precise temporal control and high ablation efficacy. Although previous studies have attempted to induce target cell

ablation during fetal development or utilized apoptotic pathways for cell ablation in adult cells, this is the first study using these approaches in combination.

We demonstrate the safety, speed, and efficiency of the inducible caspase 9 system through a unique validation approach. Furthermore, this approach has potential applicability in kidney failure models that lack effective animal models, as well as in kidney regeneration, a field we have previously explored. We believe that the combination of apoptosis control with targeted cell ablation in the fetus, which has been successfully applied in Cre-loxP and rat models, holds great potential for application in other organs and animal models.

Below is a summary of these important strengths of our model, presented in relevant sections of the manuscript:

1. The speed of ablation is faster in our model compared with the DTA model (**Fig. 3e, f, Supplementary Fig. 7**).
2. The ablation efficiency reaches 94% (**Fig. 3d**) and is 100% when used in combination with AT406 (**Fig. 4j**).
3. Regarding fetal cell ablation, our model is superior to both the DTR model, which requires direct intrauterine injection, and the DTA model, which requires the administration of highly toxic tamoxifen (**Table 1**). We have confirmed that CID has no adverse effects in the wild-type fetus (**Supplementary Fig. 2**) in contrast to tamoxifen, which causes damage to wild-type fetuses. These are presented in the **Results** section (**Generation of Six2-iC9 mice and *in utero* ablation of fetal NPCs in Six2-iC9^{+/+} mice**).
4. We have developed a kidney failure model with consistent severity, which is more severe than that achieved in conventional models; our approach allows a simple approach to adjust the severity of failure (**Fig. 2**).
5. Through our unique experimental system comparing solid organs with reaggregated spheres, we demonstrate the importance of iC9 expression levels in hard-to-ablate solid organs. Furthermore, we have provided the first evidence in animal models that cell death can be induced even with low iC9 expression with XIAP inhibition using AT406 (**Fig. 4**).
6. Development of Cre-loxP models and cross-species rat models is another novel aspect of our study (**Fig. 5**).

We present the chimeric application as an example of how our system can be utilized. In this study, we demonstrate the *in vivo* maturation of human cells in chimeras, which differs from the results of [Fujimoto et al \(Cell. Rep. 2020, doi: doi.org/10.1016/j.celrep.2020.108130\)](https://doi.org/10.1016/j.celrep.2020.108130), which only showed *in vitro* human–mouse chimeric kidneys and of [Taguchi \(Cell Stem Cell. 2014, doi: doi.org/10.1016/j.stem.2013.11.010\)](https://doi.org/10.1016/j.stem.2013.11.010), which only demonstrated the self-aggregation of human kidney organoids. We cannot prove whether our achievement was specifically dependent on the iC9 model. We do not claim that our approach in regenerating chimeric nephrons is superior to conventional replacement models. Whether faster ablation improves ablation efficiency remains

a topic for future investigation, which we acknowledge in the **Discussion** section as follows:

- “It is also required to compare the speed of NPC elimination between the iC9 and the DTA models, to determine its impact on replacement efficiency.”

We have also revised the **Introduction** section to avoid overemphasizing human chimeras by removing the terms “successfully” and “which represents significant progress in regenerative research” from the sentence “Using this model, we successfully developed a human–mouse chimeric kidney that matures in vivo, which represents significant progress in regenerative research.”

Additionally, we have addressed Reviewer #1’s comments regarding the application of this system in generating chimeric organs in other point-by-point responses outlined below.

1. From whole kidney views in Fig 6b, 6c, 6g, 7b, 7g, it is very unlikely that injected GFP+ cells interact with Six2-iC9 tissues extensively. The injected GFP+ cells merely seem to be proliferating around injection sites and these dividing cells might be occasionally incorporated into the CD of host mice. Did the injected GFP+ RPCs/NPCs enhanced UB branching of host kidney? Is there any other evidence for reconstitution of tissue interactions between GFP+ cells and host tissues, other than a few images of GFP+ cells next to host CK8+ cells in Fig 6i and Fig 7i?

We thank the Reviewer for their insightful inquiry. We show that the branching of ureteric buds is impaired and tips are not formed after NPC ablation (**Fig. 3b, c**). Furthermore, following the injection of mouse renal progenitor cells, ureteric bud tips form and invade the cap mesenchyme derived from the injected NPCs (**Fig. 6d, Supplementary Fig. 12b**).

We also include additional images to illustrate the interaction between the GFP+ donor cells and collecting ducts in the kidneys of Six2-iC9 mice (**Fig. 6g, Supplementary Fig. 12c**).

2. How many per a kidney did the authors identify the connections of GFP+ cells and the host mouse CD in Fig 6i and Fig 7i when how many nephrons (or glomeruli) are formed? The numbers of connections and glomeruli per a kidney should be presented. Because many nephrons are formed, these GFP+DT to CD connections should be observed frequently if GFP+ cells interact with the host mouse kidney.

We thank the Reviewer for giving us the opportunity to clarify this detail. Renal progenitor cell injection into fetal kidneys is performed through a highly delicate technique involving the injection of a suspension using a mouth pipette connected with a glass capillary. The efficiency of chimeric nephron formation in each kidney largely depends on the quantity and quality of the injected cells (Takamura. *J. Clin.*

Med. 2022, doi: 10.3390/jcm11237237). Additionally, counting the number of was more straightforward than counting connecting points between mature nephrons. Therefore, we evaluated the efficiency of chimera nephron generation based on the number of chimeric cap mesenchymes within the area of cell injection, rather than in the entire kidney, on day 4 of culture (**Fig. 6c, e**).

We have added the following sentences to the **Results** section (**Intra- and interspecies NPC replacement and nephron regeneration with CID-induced ablation of host NPCs**)

- “Since the efficiency of chimera formation depends on the distribution of injected RPCs⁴⁶, we assessed the efficiency within the areas of donor cell injection rather than the entire kidney. The frequency of the chimeric CMs consisting of the donor NPCs and the host ureteric buds regenerated to 49% of the native CMs (Fig. 6c, e).”

We have also added the following sentence to the **Discussion** section:

- “Considering that the density of regenerated CMs was about half of that of the native CMs even when utilizing allogeneic cells, improvements in the cell injection method are necessary to achieve maximum replacement⁴⁶.”

3. Do those GFP+ cells adjacent to host CK8+ cells express DT markers? ECAD is not specific to the DT. A DT-specific marker should be used replacing ECAD.

ECAD expression is extremely low in proximal tubules (Prozialeck. BMC. Physiol. 2004. doi: 10.1186/1472-6793-4-10). Additionally, recent reports from multiple groups have utilized ECAD as a distal tubule marker (Subramanian. Nat. Commun. 2019. doi: doi.org/10.1038/s41467-019-13382-0 and Low. Cell Stem Cell. 2019. doi:10.1016/j.stem.2019.06.009). Therefore, we have adopted ECAD as a marker for distal tubules in the present study.

However, since ECAD may also be expressed in collecting ducts and therefore may lack specificity, we have incorporated TFAP2B as a distal tubule-specific marker (Miao. Nat. Commun. 2021. doi.org/10.1038/s41467-021-22266-1). In mouse–mouse chimeric nephrons, we demonstrate the connection between TFAP2b⁺/GFP⁺ cells and CK8⁺/GFP⁻ cells (**Fig. 6g**).

4. Was there any occasion where GFP+ cells are incorporated adjacent to CK8-non-CD host cells? If so, the number of the occasions per a kidney should be presented.

We thank the Reviewer for giving us the opportunity to respond. We have determined the proportion of host cells mixed with the donor-derived distal tubules within randomly selected areas in immunostaining images obtained on day 7; we did not evaluate the entire kidney due to the aforementioned reason (Reviewer #1, Point 2). We have determined that 17% of the cells were host cells (**Fig. 6h**), reflecting contamination of the CM with host NPCs on day 4 (**Fig. 6f**).

5. NPC can differentiate when cultured *ex vivo* or transplanted into immunodeficient mice. What is significant advantage with the use of Six2-iC9 kidneys? Appropriate controls should be used in the study.

An advantage of regenerating human nephrons using animal fetal kidneys as a scaffold is the ability to reproduce connections with the excretory pathway *in vivo*, which cannot be achieved with organoids. Such connections do not occur when kidney organoids are transplanted into adult kidneys ([Matsui, *Commun. Biol.* 2023, doi.org/10.1038/s42003-023-05484-9](https://doi.org/10.1038/s42003-023-05484-9)). In the present study, we observe a connection between the regenerated human nephrons and the scaffold mouse collecting ducts *in vivo*, which serves as a proof of our concept. However, improving the replacement efficiency across the species barrier remains a challenge for future research, particularly in models utilizing human cells.

We have added the following sentences to the **Discussion** section:

- “Although various methods have evaluated the generation of *ex vivo* kidney organoids to mimic kidney development^{49,72,73}, these organoids do not connect to the urinary tract after transplantation and are unable to survive long-term⁷⁴.”
- “Regenerated human nephrons can connect with the scaffold mouse nephrons^{32,51,77}, and the present study demonstrates that human nephrons connected to mouse collecting ducts can be regenerated *in vivo*. Furthermore, if transplanted with an intact connection to the ureter and bladder, fetal kidneys can be anastomosed with the recipient’s urinary tract⁷⁸. Additionally, partially replacing cells in fetal organs with allogeneic components can reduce rejection³³.”

We also provide the following original statement for consideration by the Reviewer:

- “Additionally, the developmental trajectories of xenogeneic nephrons should be evaluated to improve the rate of replacement by human NPCs. Regarding species-specific differences, cellular size, developmental velocities, and paracrine factors should be evaluated.”

6. The authors already claimed that Six2CreERT2; R26RDTA system resulted in efficient ablation of nephron progenitor cells and that human iPSC-derived NPCs could be successfully tested in the Six2CreERT2; R26RDTA system (Cell Reports 2020). Some minor differences were presented in Supplementary Fig. 4a, b. However, for testing human NPCs, it was unclear what was the advancement in the iC9, compared with the published Six2CreER-iDTR.

We apologize for the misunderstanding. While we have demonstrated the *in vivo* regeneration of chimeric human nephrons, we have been unable to confirm that the faster elimination speed directly translates to superior replacement efficiency.

Nevertheless, our iC9 model has several advantages over conventional models, as

mentioned above, and the different cell ablation models will be valuable in examining the relationship between the elimination speed and the replacement efficiency in future studies.

To address this concern, we have added the following sentence to the **Discussion** section:

- “It is also required to compare the speed of NPC elimination between the iC9 and the DTA models, to determine its impact on replacement efficiency.”

Reviewer #2

1. In Result 1, the authors aimed to generate Six2-iC9 mice. However, it is unclear why chose to test the CAG-iC9 vector in porcine fibroblasts instead of conducting relevant experiments in mouse fibroblasts. It is recommended that relevant experiments should be performed in mouse cells to provide a more coherent progression.

We thank the Reviewer for their valid recommendation. We envision generating a human–pig chimera using a pig ablation model in future studies; therefore, we have validated the iC9 vector using porcine fibroblasts. We have also confirmed in mouse embryonic fibroblasts that ablation occurs with the addition of CID. These additional experiments are presented in the new **Supplementary Fig. 1b**.

2. In Supplementary Fig. 1b, it is essential for the authors to offer a comprehensive explanation regarding the presence of a band around 318 bp in the wild-type sample and the absence of a 3200 bp band in the Six2-iC9^{+/-} (heterozygous) sample. A detailed clarification needs to be provided.

The band was 3200 bp in the knock-in allele and 420 bp in the wild-type allele. In heterozygous mice, the smaller wild-type allele was preferentially amplified, hindering our ability to detect the knock-in band. Therefore, an additional PCR was necessary to specifically confirm the presence of the knock-in allele.

To clarify the interpretation of the genotyping results, we have added the following explanation to the **Results** section (**Generation of Six2-iC9 mice and *in utero* ablation of fetal NPCs in Six2-iC9^{+/+} mice**).

- “Using primers targeting the *Six2* sequence flanking the knock-in region (*Six2* 5' and 3' arms), the knock-in allele generates a 3200-bp PCR product, which is significantly larger than the 420-bp product from the wild-type allele. In Six2-iC9^{+/-} mice, amplification of the knock-in allele is inhibited; therefore, an additional PCR targeting the 5' knock-in site was performed to distinguish Six2-iC9^{+/-} mice from the wild-type mice.”

3. In Supplementary Fig. 2, the authors assessed the toxicity of CID on fetuses by evaluating the survival rate of E18.5 fetuses through cesarean section. For a

more rigorous assessment, it would be beneficial to examine the postnatal survival rate of the mice after birth.

We thank the Reviewer for their helpful recommendation. Accordingly, we evaluated survival in 2-month-old wild-type mice treated with CID. We observed a survival rate of 100% in both the untreated and treated groups (13/13 and 9/9, respectively), with no significant differences in body weight between the two groups (**Supplementary Fig. 2c**).

4. In Fig. 2a, the samples size of biological replicates for measuring Serum creatinine (sCr), blood urea nitrogen (BUN), and urine albumin-to-creatinine ratio (ACR) in 1 month or 2 months is inconsistent. Can the authors elucidate this inconsistency? The sample size (n) should be provided in the corresponding figure legends.

Due to the death of one mouse after 1 month of age, the number of surviving mice decreased from six at 1 month of age to five at 2 months of age. All mice had been included in the analysis. However, to evaluate the uniformity and sex differences in this CKD model, we generated a new cohort of 21 Six2-iC9^{+/+} mice via *in vitro* fertilization, which were administered CID, for further analyses (**Fig. 2a**). All data are documented in **Supplementary Data**, and the *in vitro* fertilization procedure has also been described in **Methods**.

5. Do clinical phenotypes of chronic kidney disease (CKD) exhibit cyst formation, a decrease in glomerular numbers, or cortical proximal tubule dilatation? To highlight the significance of utilizing the ablation system for constructing animal disease models, a comprehensive and detailed comparison between the CKD model and clinical phenotypes should be provided, elucidating the similarities and differences between them and emphasizing the importance and implications of the model for studying CKD.

We thank the Reviewer for their insightful comments. The following statements have been added to the **Discussion** section:

- “Nephron loss is a common pathophysiologic feature associated with low birthweight, aging, and hypertension, which imposes hemodynamic and metabolic overload on remaining nephrons⁵⁵. The imposed burden placed by nephron loss stimulates signaling including PPAR α , AMPK, and mTORC1, leading to tubular hypertrophy and cyst formation, thereby contributing to CKD progression⁵⁶. Indeed, we observed the characteristic CKD features, including glomerular hypertrophy and sclerosis, tubular dilation and atrophy, interstitial fibrosis, and inflammatory cell infiltration, in this model⁵⁷.”
- “The detailed mechanisms underlying the progression from nephron loss to kidney injury remain incompletely understood, and our model should serve as a valuable experimental tool for future investigations on potential therapeutic targets.”

6. Fig 2i showed immunostaining of Six2-iC9 mice treated with CID. Proper controls, including Six2-iC9 mice without CID treatment and wild-type mice, should be performed.

We apologize for the misunderstanding. We had originally included Six2-iC9^{+/+} mice without CID treatment as the control group in **Fig. 2i**. Additionally, based on the Reviewer's suggestion, we ensured an adequate number ($n = 8-9$) of wild-type individuals without CID treatment and redefined this as the negative control (**Fig. 2**).

Regarding Six2-iC9^{+/+} mice without CID treatment, as mentioned in response to Point 7 by Reviewer #2, we have confirmed that these mice were comparable to the wild-type mice and that unexpected NPC ablation was not present in these mice (**Fig. 1h, I, Supplementary Fig. 3, Fig. 3d, Supplementary Fig. 6**).

7. There is a lack of wild-type controls in Fig. 1i, 1j, and Fig. 3. The expression level of Six2 can provide insights about whether the Caspase 9-induced apoptosis system is due to leakage. Therefore, it is crucial to conduct a detailed evaluation of Six2 expression levels in both Six2-iC9 mice with or without CID treatment and wild-type mice.

We have added wild-type B6 mice without CID treatment as a control and confirmed that there is no unexpected activation of iC9 (**Fig. 1h, I, Supplementary Fig. 3, Fig. 3d, Supplementary Fig. 6**).

8. The authors stated that the addition of the XIAP inhibitor lowers the threshold level of iC9 expression. The extent of threshold reduction should be quantified, for example, by comparing the level of NPC ablation in Six2-iC9^{+/+} mice treated with CID only and Six2-iC9^{+/-} mice treated with both AT406 and CID.

Thanks to the Reviewer's suggestion, we have increased the AT406 concentration from 10 μ M to 100 μ M, achieving an NPC ablation rate of 100%, confirming that the threshold-lowering effect of AT406 was sufficient to offset the low iC9 expression levels in our model (**Fig. 4j**).

9. The iC9 system used in the study has not been improved compared to previous versions. As a result, the efficiency of cell removal in adult rodent models still remains unsatisfactory and unavoidable. How can you ensure the efficient achievement of complete organ ablation? To achieve more efficient whole-organ ablation, optimization can be explored in terms of dimerization-based ablation systems (<https://doi.org/10.1038/s41589-023-01281-x>). Based on this, a brief discussion can be conducted.

We agree with the Reviewer that our results do not imply that sufficient ablation has been achieved in adult organs. We have added the following statement to the **Discussion**, with reference to the article suggested by the Reviewer:

- "Third, it remains unclear whether adult organs, especially those with robust compensatory proliferation such as the liver⁴, can be targeted for ablation. We aim to verify this aspect in models that use other promoters or in those created

by breeding with the ROSA26-iC9 mice with the appropriate Cre mouse lines. As demonstrated in the present study, increasing iC9 expression and the combined use of XIAP inhibition is a promising approach to increase ablation efficacy. Furthermore, combining genetic engineering strategies, such as selecting appropriate promoters, utilizing enhancers, or employing innovative systems like combinatorial protein dimerization to sustain high gene expression while minimizing leakage⁸³, can be explored.”

10. The results in line 221-224 is inconsistent with the figures. The corresponding Fig. 4b, 4c, and 4d should actually refer to Fig. 4h, 4i, and 4j, respectively.

We thank the Reviewer for their careful observation. We have corrected the figure numbers.

11. The authors employed a bacterial artificial chromosome (BAC) transgenic approach to introduce a donor vector containing iC9 linked with tdTomato under the mouse *Six2* gene. Why not use the rat's *Six2* promoter? In addition, targeted gene knock-in can provide more stable integration and expression of the desired transgene. Why not insert iC9 linked with tdTomato specifically after the last exon of the rat's *Six2* gene? Please provide necessary explanation for this.

We utilized the BAC approach due to the following reasons, which are added to the **Discussion** section:

1. The donor vector copy numbers are higher in transgenic animals than in knock-in animals, which is an advantage, given that higher iC9 expression levels are beneficial for apoptosis induction. Up to 30 copies were present in the rats analyzed in the present study.
2. Transgenic lines can be established more quickly compared with the knock-in model.
3. Validation in *Six2*-iC9 mice had already confirmed that the mouse *Six2* promoter is capable of sufficiently driving iC9 expression.

12. In Figure 6g-i, the authors indicate rat cell integration to fetal *Six2*-iC9^{+/+} kidney by immunohistochemistry. The chimeric efficiency should be provided using flow cytometry.

We thank the Reviewer for their suggestion. While flow cytometry is a powerful method to calculate cell ratios, we did not implement this approach, because successfully dissociating whole chimeric fetal kidneys into single-cell suspensions while maintaining minimal cell damage during processing is challenging, hindering the ability to accurately determine chimeric efficacy.

Reviewer #3

1. Have the authors assessed how consistent was the ablation of NPCs and the reduction of mature glomeruli within the CID exposed fetuses? The graphs in Fig

1 h and i show 3-5 mice per group whereas an average litter should be closer to 8 mice. Have they assessed whether mouse gender affected the degree of ablation? The reviewer has similar concerns for Figure 5.

We thank the Reviewer for their inquiry. The evaluation of neonatal mice (**Fig. 1**) was conducted using randomly selected neonates without regard to sex, whereas the remaining littermates were allowed to age for additional analysis after maturation.

We demonstrate the consistency of NPC ablation based on the uniform severity of kidney injury at 1 and 2 months of age and the uniform decrease in the number of glomeruli at 2 months of age (**Fig. 2a, d**).

Additionally, we have confirmed the absence of sex-related variability in NPC ablation efficiency in our assessments of 1- and 2-month-old mice (**Fig. 2a**).

We have also confirmed that there is no sex difference in the extent of NPC ablation in the neonatal stage in Six2-iC9 rats (**Fig. 5i**).

2. Could the authors provide more details in the variability of the CKD model described in Figure 2? While they started with 18 mice (mentioned in results), they report the functional measurements for 5-6 mice per group. The pathology analysis is also limited to 3 mice. It would be important to include the phenotypes of the entire cohort to get a better idea of the potential variability.

As indicated in our response to Reviewer #2 (Point 4), we generated a new cohort of 21 Six2-iC9^{+/+} mice via *in vitro* fertilization, which were administered CID, and confirmed the consistent presence of severe kidney injury based on blood and urine tests at 1 and 2 months of age, as well as RT-PCR results at 2 months of age (**Fig. 2**). As indicated in the **Supplementary Data**, we have randomly reselected four mice from each group for pathologic evaluation. All data are documented in **Supplementary Data**, and the *in vitro* fertilization procedure has also been described in **Methods**.

3. To better characterize the CKD phenotype, it would be helpful to include RT-PCR analysis of pro-fibrotic genes. Was there inflammatory cell infiltration along with the fibrosis?

We show that the levels of fibrosis markers *Col1a1*, *Fn1*, and *Acta2* are significantly elevated in our CKD model, as determined by RT-PCR of kidneys harvested from 2-month-old mice (**Fig. 2k**). We also demonstrate that their expression levels are associated with serum creatinine levels (**Supplementary Fig. 4d**). Additionally, we include immunostaining for IBA1 to demonstrate interstitial macrophage infiltration (**Fig. 2f**).

4. Since CID administration in Six2-iC9 neonates leads to ablation of NPCs (SuppFig 5), have the authors examined if these mice develop CKD? Probably there is a typo in the results, Six2-iC9^{+/-} instead of Six2-iC9^{+/+}. Likewise, the legend of Supp Fig 6 mentions homozygous instead of heterozygous.

We thank the Reviewer for their careful observation and apologize for our oversight. We have corrected the two errors.

We have also demonstrated that the degree of kidney injury is much milder when CID is administered to Six2-iC9^{+/+} neonates, compared to E13.5 fetuses (**Supplementary Fig. 5d**).

Point-by-point Responses to the Reviewers' Comments

Date: January 31, 2025

Journal: *Nature Communications*

Manuscript ID: NCOMMS-24-21286B

Manuscript title: Caspase 9-induced apoptosis enables efficient fetal cell ablation and disease modeling

Authors: Kenji Matsui, Masahito Watanabe, Shutaro Yamamoto, Shiho Kawagoe, Takumi Ikeda, Hinari Ohashi, Takafumi Kuroda, Nagisa Koda, Keita Morimoto, Yoshitaka Kinoshita, Yuka Inage, Yatsumu Saito, Shohei Fukunaga, Toshinari Fujimoto, Susumu Tajiri, Kei Matsumoto, Eiji Kobayashi, Takashi Yokoo and Shuichiro Yamanaka

Responses to the reviewers' comments

We greatly appreciate the reviewers' detailed comments and suggestions. We have addressed all editorial requirements outlined in the Author Checklist. Below are our point-by-point responses to the reviewers' comments.

Reviewer #1

There is no response addressing my concern regarding the significance or novelty of the current study. It has been well established over the past decade that transplantation of kidney organoids promotes maturation, which is not a novel finding of this study. For years, the authors have already demonstrated that chimeric fetal kidneys with NPC are effective and useful.

Moreover, the advancement claimed in the current study is not definitive. To substantiate the claimed progress over their previous publications, the authors should include a comparison using DTA as a negative control. Without proper side-by-side controls, no definitive conclusions can be drawn. Addressing this issue requires extensive experimentation rather than mere editorial changes or minor experiments.

> Thank you for your valuable feedback and for highlighting this important issue. We recognize its significance and will address it as a key aspect in future studies.